# In vitro approaches to study centriole and cilium function in early mouse embryogenesis

Isabella Voelkl[1,2] , Tamara Civetta[1] , Mirijam Egg[1], Marie Huber[1] , Songjie Feng[1,2] , Alexander Dammermann[1] , Christa Buecker[1]

Although centrioles and primary cilia play an essential role in early mammalian development, their specific function during the interval between their initial formation and the subsequent arrest of embryogenesis in embryos deficient in centrioles or cilia remains largely unexplored. Here, we demonstrate that different 3D in vitro model systems recapitulate early centriole and cilium formation in mouse development. Centrioles and cilia are dispensable in 3D in vitro mouse rosettes, a model system that mimics key events of implantation, including polarization and lumenogenesis. In gastruloids, a model system that recapitulates developmental processes up to E8.5, centriole loss results in early disassembly that can be rescued by additional p53 deletion. In contrast, cells devoid of cilia continue to form elongated, differentiated and polarized gastruloids, with minor differences at 96 h. Finally, we show that in a mutant affecting the centriolar distal appendages, cilia are absent from 2D cultures, but are capable of forming in 3D rosettes and gastruloids, highlighting the importance of multifactorial 3D environment setups in developmental studies.

## Introduction

Centriole-based centrosomes serve as the predominant microtubule organizing centers in animal cells, playing a vital role in mitotic spindle assembly and cell division (Nigg & Raff, 2009; Bornens, 2012). In addition, centrioles also provide the structural foundation of primary cilia—solitary, antenna-like sensory organelles that project from the surface of most vertebrate cells to mediate cellular signaling (Sorokin, 1968). Cilium formation requires the gradual maturation of centrioles over two cell cycles, enabling the mother centriole to transform into the basal body (Kong et al, 2014; Xiao et al, 2020 Preprint). In mammals, this process depends on the distal appendage protein CEP83, which facilitates the docking of the basal body to the plasma membrane

(Tanos et al, 2013; Lo et al, 2019). This connection then serves as a structural template for the extension of the axoneme, the microtubule-based core of the cilium, with the assistance of the intraflagellar transport (IFT) machinery, including the IFT-B core component IFT88 (Haycraft et al, 2007). Although the ciliary membrane comprises only around 1/200[th] of the total cell surface area, it is highly enriched in receptors, channels and effectors, essential for homeostasis and tissue patterning during embryogenesis. These components are crucial for mediating signal transduction, including pathways such as Hedgehog (Hh) and Wnt signaling, in particular during development (Mill et al, 2023). Consequently, dysfunction of primary cilia can give rise to pleiotropic and sometimes severe disorders, collectively termed ciliopathies, which can affect multiple organs, including the kidneys, eyes, liver, brain, heart, lung and skeleton (Reiter & Leroux, 2017). To understand how centrioles and primary cilia develop and acquire their function during mouse embryonic development, it is essential to dissect their emergence within the embryo.

In contrast to human, Caenorhabditis elegans and zebrafish development, in which centrioles are indispensable (Pelletier et al, 2006; Yabe et al, 2007; Avidor-Reiss et al, 2019), the initial cell divisions of a developing mouse embryo after fertilization occur in the absence of centrioles, as both the oocyte and the sperm undergo centriole degeneration (Gueth-Hallonet et al, 1993; Manandhar et al, 1999). The first acentriolar foci of pericentriolar material (PCM) appear at the morula stage on embryonic day E2.5, and centrioles are formed de novo slightly later at the blastocyst stage at E3.5 (Howe & FitzHarris, 2013). Primary cilia emerge even later and are first detected on epiblast cells after implantation, around E5.5-E6, coinciding with cavitation and the onset of gastrulation. By E6, over 30% of epiblast cells are ciliated, and by E8, primary cilia are present on cells of all three germ layers, including the node (Bangs et al, 2015), the site where the left-right body axis is established (Brennan et al, 2002).

Centriole and cilium loss are lethal in mouse embryos. Homozygous Plk4 KO mouse embryos lacking centrioles arrest in development at E7.5, accompanied by delays in cell division and ultimately apoptosis (Hudson et al, 2001). Mutations in IFT proteins

[1]Max Perutz Labs, University of Vienna, Vienna BioCenter (VBC), Vienna, Austria    [2]Vienna BioCenter PhD Program, Doctoral School of the University of Vienna and Medical University of Vienna, Vienna, Austria

Correspondence: christa.buecker@univie.ac.at

inevitably result in mouse embryonic lethality by mid-gestation, around E11 (Murcia et al, 2000) because of the disruption of essential signaling pathways such as Wnt and Hh (Huangfu et al, 2003; Cortellino et al, 2009). Although primary cilia play essential roles in early mammalian development (Gerdes et al, 2009; Bangs & Anderson, 2017; Amack, 2022), their specific functions during the interval between their initial formation and the subsequent arrest of embryogenesis in cilia-deficient embryos remain largely unexplored. This limited understanding stems from the challenges associated with studying primary ciliogenesis in vivo, as these critical stages of development are difficult to observe in situ in the living embryo, offering only a limited, static perspective of developmental processes.

Mouse embryonic stem cells (mESCs) are a well-established in vitro model to study different aspects of mouse embryonic development. They are derived from the inner cell mass of the embryo, molecularly resembling the preimplantation epiblast (Nichols & Smith, 2009) and can be differentiated into formative, epiblast-like cells (EpiLCs) (Hayashi et al, 2011; Buecker et al, 2014). mESCs serve as a powerful tool to generate various 3D in vitro model systems, including the embedded rosette model, which mimics critical aspects of implantation such as polarization and lumenogenesis (Bedzhov & Zernicka-Goetz, 2014; Shahbazi et al, 2017), and gastruloids, a developmental model system which recapitulates early embryonic cell fate decisions up to ~8.5 d post-fertilization (Beccari et al, 2018).

In this study, we used 3D in vitro rosettes and mouse gastruloids as in vitro model systems to investigate the formation and function of centrioles and cilia during early mouse embryonic development. We show that differentiation per se is not a driver of cilium formation during transition of mESCs to EpiLCs; only in combination with polarization and lumenogenesis is ciliogenesis enhanced. Cilium- (Ift88 KO, Cep83 KO) and centriole (Plk4 KO)-deficient rosettes maintain rosette-like morphology with a central lumen. However, Plk4 is indispensable for gastruloid formation, as its loss leads to gradual disassembly in homozygous mutants, a phenotype that can be rescued by additional p53 deletion. In contrast, Cep83 and Ift88-mutant mESCs develop into elongated gastruloids, with minor morphological differences mainly at 96 h. Finally, a CEP83 truncation mutant (Cep83Δexon4) reveals surprising differential phenotypes between 2D cultures and 3D gastruloids, highlighting the importance of 3D models in developmental studies. This study provides the first comprehensive analysis of primary ciliogenesis and centriole formation in in vitro differentiation models of early development.

## Results

### Primary ciliogenesis during exit from naive pluripotency

In the mouse embryo, cilia first arise on epiblast cells after implantation at the time of cavitation at E5.5-E6. In contrast, mESCs already possess the capacity to form primary cilia, which are found on a small proportion (~5%) of cells (Bangs et al, 2015); however, it remains unclear whether cell differentiation enhances ciliogenesis in vitro. Removal of 2iLIF irreversibly differentiates naive mESCs

into epiblast-like cells (EpiLCs) (Hayashi et al, 2011; Buecker et al, 2014), a process described as transitioning into the formative state of pluripotency or the exit from naive pluripotency (Smith, 2017). Here, we tested the potential of primary cilia to form during the transition of naive mESCs to formative EpiLCs (Fig 1A) using a previously established, highly efficient and reproducible protocol (Buecker et al, 2014; Thomas et al, 2021; Romeike et al, 2022; Boileau et al, 2023; Schulz et al, 2024). We differentiated mESCs into EpiLCs for 48 h by removal of 2iLIF and determined their ciliation rate based on immunofluorescence staining using antibodies against the ciliary marker ARL13B and acetylated α-tubulin, a marker for stable tubulin (Fig 1B). Under mESC conditions, ~5% of cells exhibited a cilium, in line with previous reports. This ciliation rate remained unchanged during the differentiation into EpiLCs, indicating that the transition into formative pluripotency itself is not a driver of ciliogenesis (Fig 1C). The length of primary cilia of both mESCs and EpiLCs was between 1 and 2 µm, with EpiLCs exhibiting moderately shorter cilia. In summary, differentiation of mESCs towards EpiLCs does not increase ciliogenesis.

Ciliogenesis increases in response to serum starvation in various cell types such as retinal pigmented epithelium cells and human embryonic kidney cells (Takahashi et al, 2018). Since differentiation did not increase the rate of ciliogenesis, we investigated whether serum starvation could enhance ciliogenesis in our mESC model. We cultivated mESCs for 24 h and removed serum and additives (Methods, Cell starvation) for an additional 24 h before assessing cilium formation via immunofluorescence microscopy (Fig 1D). Starvation itself was monitored by expression of p27 (Fig 1E), a cyclin-dependent kinase inhibitor that plays a crucial role in regulating the cell cycle and maintaining cellular quiescence. Elevated p27 levels are often associated with cell cycle arrest, particularly during starvation-induced quiescence (Li et al, 2019). Serum-deprived mESCs significantly increased ciliation compared with non-starved cells whereas cilium length remained unaffected (Fig 1F). As p27 has been shown to directly repress the pluripotency gene Sox2 during embryonic stem cell differentiation (Li et al, 2012), we tested if starvation alters the naive pluripotent state of mESCs. We conducted RT-qPCRs against naive (Tbx3, Klf4, Esrrb), core (Oct4, Sox2), and formative pluripotency markers (Fgf5, Oct6, Otx2) (Fig S1). Our data showed no difference between 24 or 48 h starved cells and the non-starved control.

Together, these results suggest that a low percentage (5%) of naive mESCs is capable of forming cilia. The efficiency of ciliogenesis is not elevated during cell differentiation of mESCs to EpiLCs but can be increased by starvation, similar to observations in other cell types.

### 3D in vitro rosettes as a model system for primary ciliogenesis

In vivo, cilia first arise on embryonic epiblast cells after implantation at the time of cavitation at E5.5-E6 (Bangs et al, 2015). During these developmental steps, the cells undergo transcriptional and morphological changes, such as polarization. Whereas EpiLCs are an excellent model system to study the transcriptional changes associated with exit from naive pluripotency, they do not recapitulate the morphological transformations occurring in vivo. Implantation itself is not tractable in vivo; therefore, we set out to

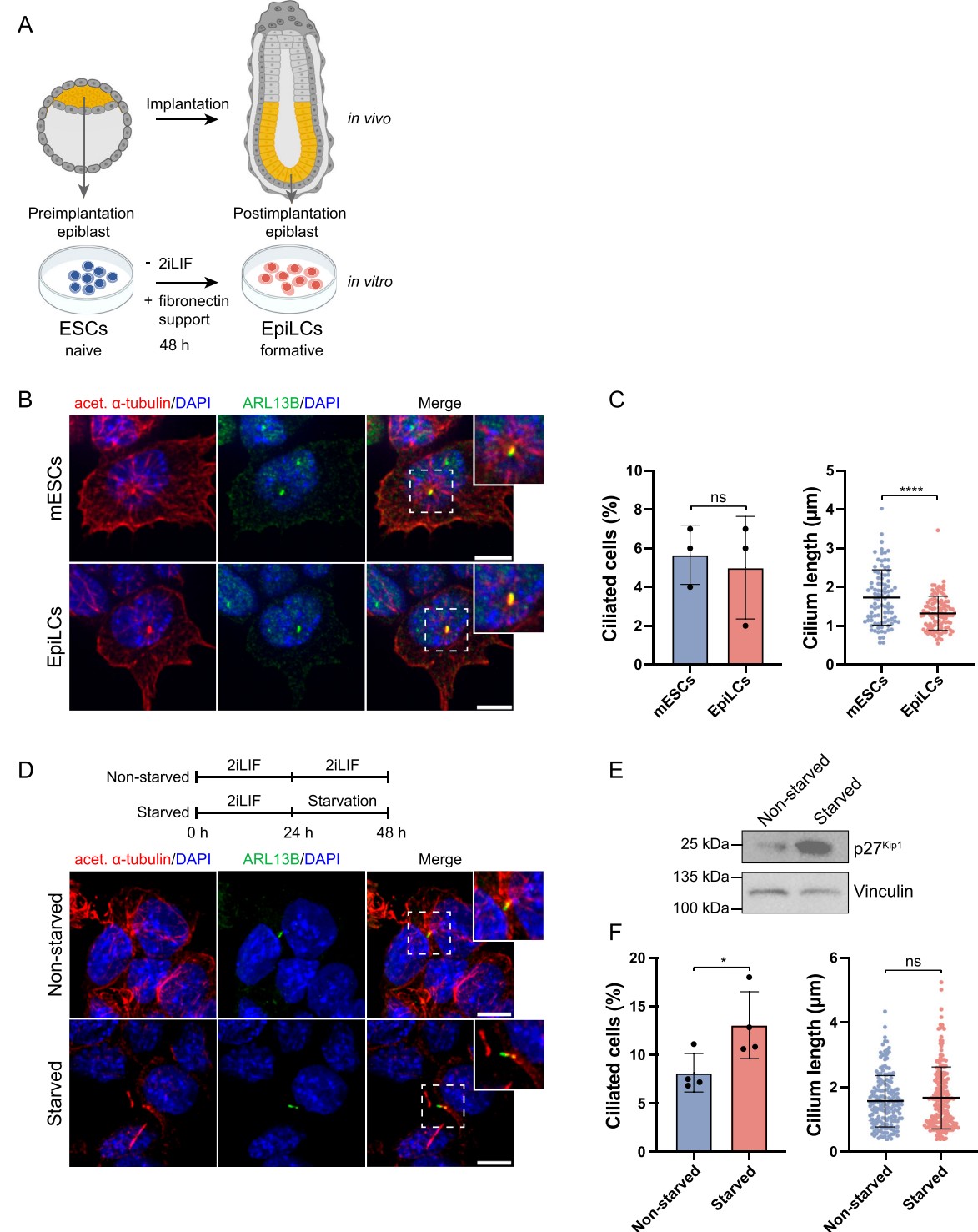

**Figure 1. Exit of naive pluripotency does not increase primary ciliogenesis.**
**(A)** Schematic overview of ESC to EpiLC transition. Upper panel depicts the corresponding cell fate decisions in vivo; lower panel the conditions for the exit of the naive state to formative EpiLCs in vitro. **(B)** Representative examples of immunofluorescence staining of naive Mouse embryonic stem cells and formative EpiLCs after 48 h of differentiation labeled with antibodies against acetylated α-tubulin (red), the ciliary marker ARL13B (green) and DAPI staining (blue). Maximum intensity projection of central z-planes. Data represent three independent experiments (n = 3), each comprising >30 images per condition, for a total of >90 images per condition. **(C)** Ciliated cells (%) and cilium length ($\mu$m) are indicated. Data represent three independent experiments (n = 3), each comprising >30 images per condition, for a total of >90 images per condition (ns, non-significant, $^{ns}P$ > 0.05, ****$P$ < 0.0001, unpaired $t$ test). **(D)** Representative examples of immunofluorescence staining of naive control Mouse embryonic stem cells (non-starved) and after 24 h serum starvation, labeled with antibodies against acetylated α-tubulin (red), ARL13B (green), and DAPI staining (blue). Maximum intensity projection of central z-planes. Data represent four independent experiments (n = 4), comprising a total of >125 images per condition (each image capturing multiple cells). **(E)** Western blot analysis of starvation marker p27 in naive control ESCs and after 24 h starvation. Vinculin was used as a loading control.

find a suitable in vitro model to study early mammalian ciliogenesis in a controlled and monitored environment. We adapted the previously published 3D in vitro rosette assay (Bedzhov & Zernicka-Goetz, 2014; Shahbazi et al, 2017), a model for polarization and lumen formation, as a system to study ciliogenesis in early mouse embryonic development. In the 3D in vitro rosette assay, mESCs were embedded in Basement Membrane Extract (BME) without 2iLIF for up to 72 h to induce cell differentiation and lumenogenesis (Fig 2A). Under 2iLIF conditions, lumenogenesis was inhibited in 3D rosettes growing for 72 h, as evidenced by the absence of the luminal marker PODXL (Fig 2B). In contrast, removal of 2iLIF promoted rosette-shaped organization of cells and lumenogenesis after 72 h (Fig 2B). We quantified the timing of lumenogenesis and evaluated the efficiency of rosette formation under differentiation conditions to assess the robustness of this model system. After 48 h, 56% of rosettes exhibited lumen formation, increasing to 100% by 72 h (Fig 2C), demonstrating the system's high reproducibility and efficiency. Investigating the suitability of this system to study ciliogenesis, we immunostained rosettes after 72 h with antibodies against ARL13B and acetylated α-tubulin (Fig 2D). Both markers revealed prominent rod-shaped primary cilia, protruding into the lumen of the rosettes. Quantifying the number of cells within the rosette exhibiting a cilium at 48 h revealed 42% of cells with a cilium. By 72 h, the proportion of ciliated cells nearly doubled such that almost every cell framing the lumen displayed a cilium. In summary, the 3D in vitro rosette system is ideally suited for investigating early ciliogenesis during mouse embryonic development in vitro, integrating key developmental processes such as cell polarization, lumenogenesis, and the dynamics of cilium assembly.

## Depletion of cilia and centrioles in mESCs to investigate early centriolar/ciliary functions

Next, we sought to investigate the role of centrioles and cilia in early mouse embryonic development. Using CRISPR/Cas9, we deleted the ciliary protein Intraflagellar Transport 88 (IFT88) (Fig 3A), a component of the intraflagellar transport machinery essential for cilium assembly and function (Pazour et al, 2000). Additionally, we targeted centrioles by deleting Polo-Like Kinase 4 (PLK4), a serine/threonine kinase with a critical role in regulating centriole duplication (Bettencourt-Dias et al, 2005; Habedanck et al, 2005; Swallow et al, 2005). In our KO approach, we designed two gRNAs for the *Ift88* KO, targeting exon 7 and the following intron, and two gRNAs for deletion of *Plk4*, specific to exon 5 and the following intron (Fig 3B). After genotyping (Fig S2A and B), we confirmed the absence of IFT88 protein in *Ift88* KO mESCs by Western blotting (Fig S2C). However, endogenous PLK4 expression was too low to be reliably detected, even in the WT condition. We therefore validated the loss of centrioles and cilia in both *Ift88* and *Plk4* KO mESCs by immunofluorescence staining, using antibodies against the ciliary marker ARL13B and the PCM marker γ-tubulin (Fig 3C). All tested *Ift88* KO clones exhibited a

complete loss of ARL13B signal. Whereas ~5% of WT cells displayed a primary cilium, none of the *Ift88* or *Plk4* KO cells analyzed exhibited cilia based on ARL13B staining (Fig 3D). Consistent with their differential effect on centriole assembly, γ-tubulin staining confirmed the presence of centrioles in all WT cells as well as in *Ift88* KO cells, whereas no centrioles were detected in *Plk4* KO cells (Fig 3D). On this basis, we designated *Ift88* KO clones as cilium KO and *Plk4* KO clones as centriole KO. Given PLK4's role in centriole duplication (Habedanck et al, 2005), we assessed whether its deletion affects cell proliferation. We therefore performed cell cycle analysis on all KOs for *Ift88* and *Plk4* by Propidium Iodide (PI) staining (Fig 3E). We observed no statistically significant variation between WT and KO cell lines in their cell cycle profiles. Furthermore, we conducted RT-qPCRs to exclude that the KO of *Ift88* or *Plk4* induces differentiation (Fig S3A and B). Our data show no significant difference between KO cells and the naive parental WT control.

Taken together, we generated *Ift88* and *Plk4* KO cell lines using a CRISPR-targeted approach to remove centrioles and cilia from mESCs. These cell lines were validated and displayed a cell cycle profile comparable to WT cells.

## 3D in vitro rosettes develop normally in the absence of centrioles and cilia

Polarization and lumenogenesis are key developmental processes during implantation associated with re-organization of the epiblast (Kim et al, 2021). To study how loss of centrioles or cilia affects these aspects, we generated 3D in vitro rosettes derived from either *Ift88* or *Plk4* KO clones. As expected, *Ift88* KO clones presented centrioles (γ-tubulin) but no cilia (ARL13B), whereas all *Plk4* KO clones showed neither centrioles nor cilia (Fig 4A). In contrast, ~80% of cells in each rosette displayed a primary cilium in the WT condition, indicated by ARL13B staining. Similarly, quantification of γ-tubulin staining confirmed the presence of centrioles in all WT and *Ift88* KO cells per rosette, whereas no centrioles were detected in *Plk4* KO cells (Fig 4B). All *Ift88* and *Plk4* KO cell lines formed organized rosettes with a central lumen after 72 h of 3D rosette formation (Fig 4C). We quantified lumen and rosette size based on masking the central z-plane (one image per rosette), using PODXL as a lumen marker and DAPI to assess rosette size. Measurements of lumen area, rosette area and the lumen-to-rosette area ratio did not show significant differences between WT and *Ift88* or *Plk4* KO rosettes (Fig 4D). These findings suggest that centrioles and cilia are dispensable for polarization and lumenogenesis during in vitro 3D differentiation of early mouse development.

## Gastruloids recapitulate ciliogenesis in vitro

*Plk4* KO mouse embryos lack centrioles and arrest development at E7.5, characterized by a delay in cell division and apoptosis (Hudson et al, 2001). Embryos lacking cilia generally do not survive beyond mid-gestation, around E11, largely due to failure of

**(F)** Ciliated cells (%) and cilium length (μm) are indicated. Data represent four independent experiments (n = 4), comprising a total of >125 images per condition (ns, non-significant, $^{ns}P$ > 0.05, *P < 0.05, unpaired *t* test). Scale bar (B, D): 10 μm.

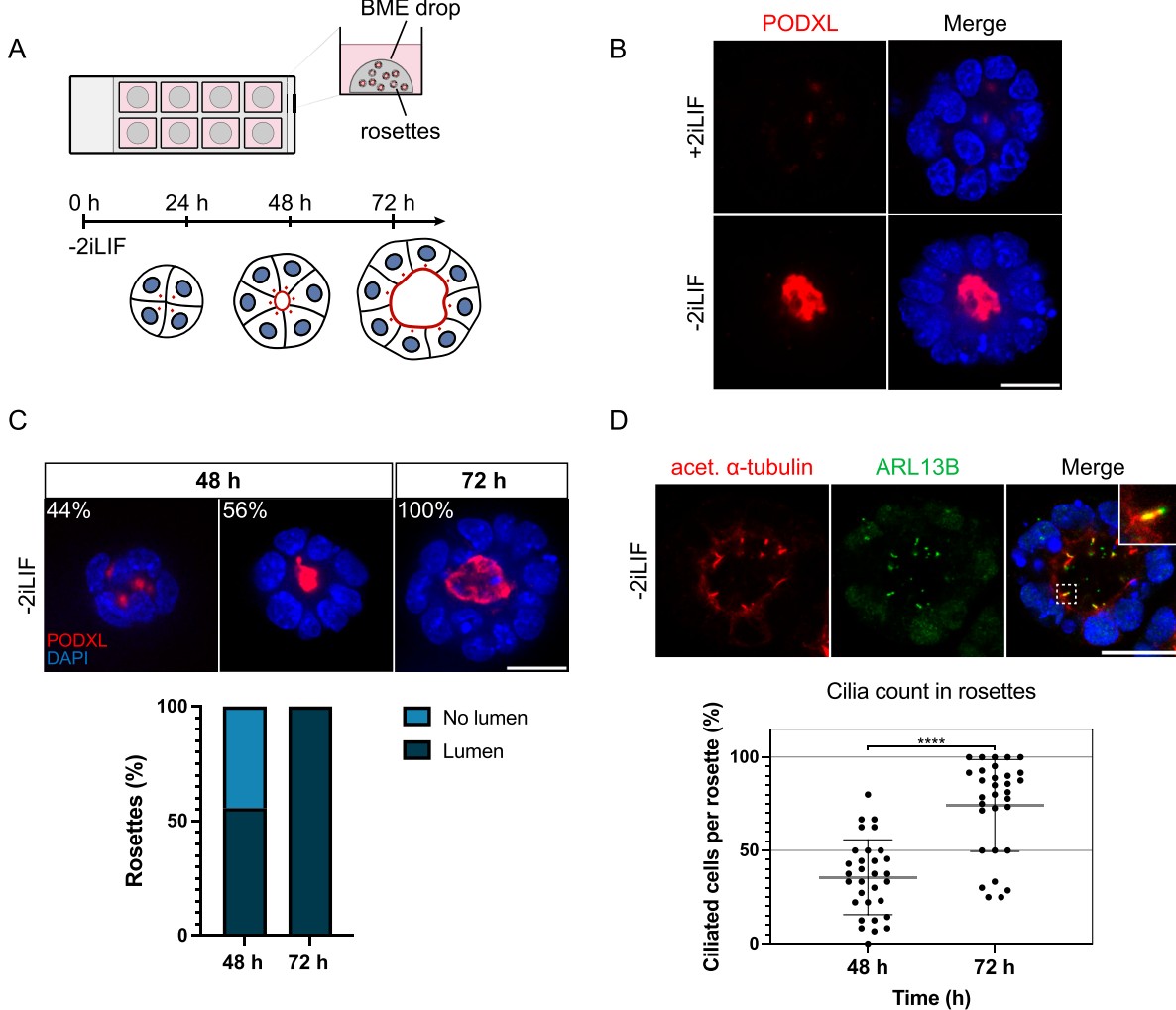

**Figure 2.  3D rosette assay recapitulates lumen and cilium formation in vitro.**
**(A)** Outline of the 3D in vitro Mouse embryonic stem cell rosette assay, based on (Bedzhov & Zernicka-Goetz, 2014; Shahbazi et al, 2017). **(B)** Immunofluorescence staining of BME-embedded 3D in vitro rosettes growing under +/− 2iLIF conditions for up to 72 h. Data represent three independent experiments (n = 3), each comprising 30 rosettes (one image per rosette) per condition, for a total of 90 rosettes per condition. Representative example of immunofluorescence staining of rosettes, labeled with antibodies against lumen marker PODXL (red) and DAPI staining (blue). The central z-plane is depicted. **(C)** Representative immunofluorescence staining and timing of lumenogenesis in Mouse embryonic stem cell-derived rosettes without 2iLIF at 48 and 72 h. Antibodies against PODXL (red) and DAPI staining (blue) are indicated. Lumen quantification of rosettes (%) from two independent experiments (n = 2), each comprising 30 rosettes (one image per rosette) per condition, for a total of 60 rosettes per condition. The central z-plane is depicted. **(D)** Immunofluorescence staining of rosettes growing without 2iLIF for 72 h, labeled with antibodies against acetylated $\alpha$-tubulin (red), ARL13B (green), and DAPI staining (blue). Maximum intensity projection of central z-planes. The number of cells exhibiting a cilium per rosette (%) was quantified in >60 rosettes (>30 rosettes/experiment measured in two independent experiments) based on ARL13B and DAPI staining (****$P$ < 0.0001, unpaired $t$ test based on % of ciliated cells per rosette of two experiments, including >30 rosettes/experiment). Each dot represents the % of cells expressing a cilium per rosette. Scale bar (B, C, D): 20 $\mu$m.

essential developmental pathways dependent on ciliary function (Cortellino et al, 2009). However, loss of PLK4 or cilia did not affect early embryonic cell state transitions and morphogenesis in vitro (Fig 4A–D). We therefore sought to investigate the role of centrioles and cilia in later stages of post-implantation development. In recent years, model systems have been developed for in vitro analysis of mammalian developmental progressions. However, to the best of our knowledge, these have not been applied to study ciliogenesis. Here, we implemented the mouse gastruloid assay, an in vitro model system that recapitulates early embryonic cell fate decisions until about 9 d after fertilization (Beccari et al, 2018), to

study cilium and centrosome biology during development. In brief, 200 mESCs per well were aggregated for 48 h in differentiation media, followed by Wnt activation using a Chiron pulse for 24 h, leading to symmetry break, anterior-posterior polarization and germ layer differentiation (Figs 5A and S4A and B). The protocol enables cultivation of gastruloids up to 120 h without shaking, corresponding up to E8.5 in vivo. We validated our system by verifying germ layer differentiation through spatiotemporal expression patterns of selected markers at different time points, ranging from 72 to 120 h (Fig S4A). The mesodermal marker Brachyury (Van Den Brink et al, 2014) was homogeneously expressed

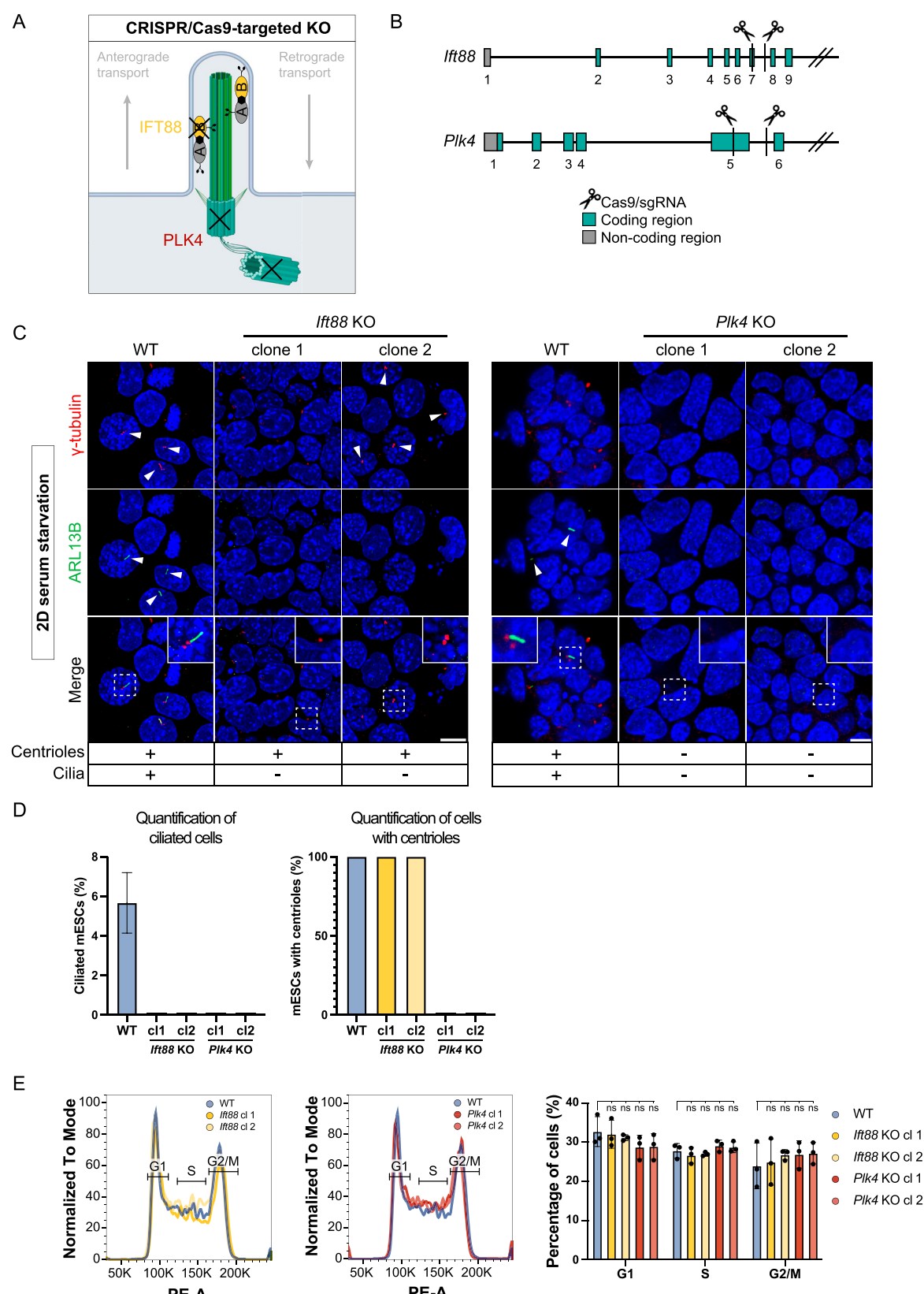

**Figure 3. CRISPR KO of cilium/centriole components *Ift88* and *Plk4* depletes cilia/centriole in 2D IF staining.**
**(A)** Schematic of a primary cilium indicating targets for CRISPR/Cas9 KOs. The cilium KO target IFT88 is a core component of the IFT-B train, while the centriole KO target PLK4 is a kinase essential for centriole duplication and hence indirectly also cilium formation. **(B)** Strategy for the generation of *Ift88* and *Plk4* KOs in Mouse embryonic stem cells. gRNAs are targeting exon 7 and the following intron in the *Ift88* KO, exon 5 and the following intron in the *Plk4* KO. Coding regions (green) and non-coding

within gastruloids up to 72 h, consistent with global Wnt activation during the Chiron pulse. By 96 h, Brachyury prominently localized to the posterior end of the gastruloids, indicating symmetry break and the establishment of anterior-posterior polarity. Additionally, gastruloids were stained for the endoderm marker SOX17 (Kanai-Azuma et al, 2002) and SOX2, a marker expressed in neuro-mesodermal progenitors (NMPs), neural progenitors, and pluripotent cells (Avilion et al, 2003; Bergsland et al, 2011; Koch et al, 2017). We observed low and unpolarized SOX2 expression at 72 h. SOX2 signal gradually increased and accumulated at the posterior end of gastruloids from 96 h on. SOX17-positive cells were detected at the posterior pole of gastruloids, visible as a one-layered cell cluster surrounding a cavity, detected in two independent experiments including a total of 31 gastruloids (one image per gastruloid) (Fig S4B). These findings are consistent with previous studies and show that our gastruloid model effectively recapitulates symmetry break, anterior-posterior polarization and germ layer differentiation in the expected time frame.

We next tested the suitability of gastruloids to study cilium formation during mammalian in vitro gastrulation. Gastruloids were fixed at 24, 48, 72, and 120 h after seeding and immunostained with antibodies against cilium and centriole markers (Fig 5B). Quantification of the mean cilia-to-centrioles ratio revealed sparse ciliation at 48 h, indicating that the majority of centrioles do not form cilia (Fig 5C). A marked increase in ciliation was observed following the Chiron pulse at 72 h, followed by a plateau at a stable level of ciliation. Ciliated epithelial cells characteristically face a lumen (van der Vaart et al, 2021); therefore, we investigated whether this also occurs in gastruloids. Gastruloids cultured for 72 h effectively recapitulated key aspects of lumen formation, indicated by the lumen marker PODXL (Fig 5D). In line with our previous 3D rosette data (Fig 2D), primary cilia were enriched at cavities and projected into these luminal compartments (Fig 5E).

Collectively, our results show that the gastruloid system is an ideal and versatile model system to implement in-depth studies of centrosome and cilia function during murine development by mimicking developmental key events including germ layer differentiation and anterior-posterior polarization.

### Cilia but not centrioles are dispensable in gastruloids

We next assessed the potential of *Plk4* and *Ift88* KO cells to form polarized, elongated mouse gastruloids. All tested *Plk4* KO clones displayed a severe phenotype, with cell aggregates progressively disassembling from the time of seeding onward, before the Chiron pulse (Fig 6A). By 72 h, the gastruloids completely degenerated and failed to progress further in development. Several studies have shown that loss of centrioles and consequently centrosomes can result in cell cycle arrest and p53 dependent apoptosis (Bazzi &

Anderson, 2014; Lambrus et al, 2015; Wong et al, 2015; Grzonka & Bazzi, 2024). We therefore investigated p53-dependence of the *Plk4* KO gastruloid phenotype by generating a *Plk4* KO-*TP53* KO cell line. After validation of the centriole and cilia depletion in the generated cell lines (Fig S5A), their potential to generate gastruloids was monitored up to 120 h. *Plk4* KO-*TP53* KO cells exhibited a morphological rescue of the *Plk4* KO phenotype and were capable of forming elongated gastruloids that were morphologically comparable to WT and *TP53* KO gastruloids for up to 120 h (Fig 6A). Our results demonstrate that the severe *Plk4* KO phenotype in gastruloids is p53-dependent, similar to what has been seen for Sas-4 (Xiao et al, 2020 *Preprint*) and Sas-6 (Grzonka & Bazzi, 2024) in vivo.

We next investigated the role of cilia in gastruloids up to 120 h. Cilium depletion in the *Ift88* KO was confirmed in gastruloids, fixed and immunostained with antibodies against ARL13B at 120 h (Fig 6B). All tested *Ift88* KO clones continued to grow elongated anterior-posterior differentiated gastruloids up to 120 h (Figs 6C and S5B). We quantified gastruloid roundness and length based on the morphology of brightfield images (Fig S5). Both *Ift88* KO clones displayed a difference in gastruloid shape particularly at 96 h, indicated by higher roundness and a decrease in length (Figs 6C and S5D). Comparison of Brachyury and SOX2 in fixed and immunostained gastruloids revealed a similar spatiotemporal distribution of both markers in WT and all tested *Ift88* KO clones at 72 and 120 h, with a marginal difference in tissue patterning of the *Ift88* KO at 96 h (Fig 6D).

In conclusion, centrioles are essential for gastruloid assembly, as their absence in *Plk4* KO clones leads to gradual gastruloid disassembly. This defect can be morphologically rescued by the additional deletion of *TP53*. In contrast, cilium-depleted *Ift88* KO clones continue to form gastruloids similar to the WT and express mesoderm and neural progenitors, indicating proper anterior-posterior polarization, albeit with minor differences in gastruloid shape at 96 h.

### 3D in vitro gastruloids, generated from *Cep83*−/− and *Cep83Δexon4* cells, continue to develop successfully

In addition to our studies with *Plk4* and *Ift88* KO cell lines in different in vitro differentiation models, we dissected the function of the distal appendage protein CEP83 in early mouse development, using gastruloids as a model system (Fig 7A). CEP83 is involved in anchoring the mother centriole to the cell membrane, a critical initiating step in mammalian ciliogenesis (Tanos et al, 2013; Lo et al, 2019; Chong et al, 2020). In contrast to *Ift88* depletion, which primarily impairs axoneme elongation, *Cep83* KO prevents the docking of the mother centriole to the plasma membrane, thereby completely abrogating ciliogenesis. This disruption also

regions (gray) are indicated. **(C)** IF-validation of centriole and cilium KOs. Representative example of immunofluorescence staining of WT, *Ift88*, and *Plk4* KO clones after induced ciliogenesis (48 h of starvation), labeled with antibodies against γ-tubulin (red), marker ARL13B (green), and DAPI staining (blue), based on >100 cells. Maximum intensity projection of central z-planes. **(C)** The images of the WT in (C) and the one from WT in Fig 7D are the same, deriving from the same experimental replicate because of both *Ift88* KO and the *Cep83Δexon4* cell line were validated using the same WT control. **(D)** Ciliated cells (%) and Mouse embryonic stem cells with centrioles (%) are indicated after induced ciliogenesis (48 h of starvation). More than 100 cells (n > 100) were quantified based on immunofluorescence staining of γ-tubulin (red), ARL13B (green), and DAPI staining (blue). **(E)** Cell cycle analysis by PI staining and flow cytometry of WT, *Ift88*, and *Plk4* KO clones, n = 3 independent experiments (ns, non-significant, $^{ns}P > 0.05$, unpaired *t* test). Scale bar (C): 10 μm.

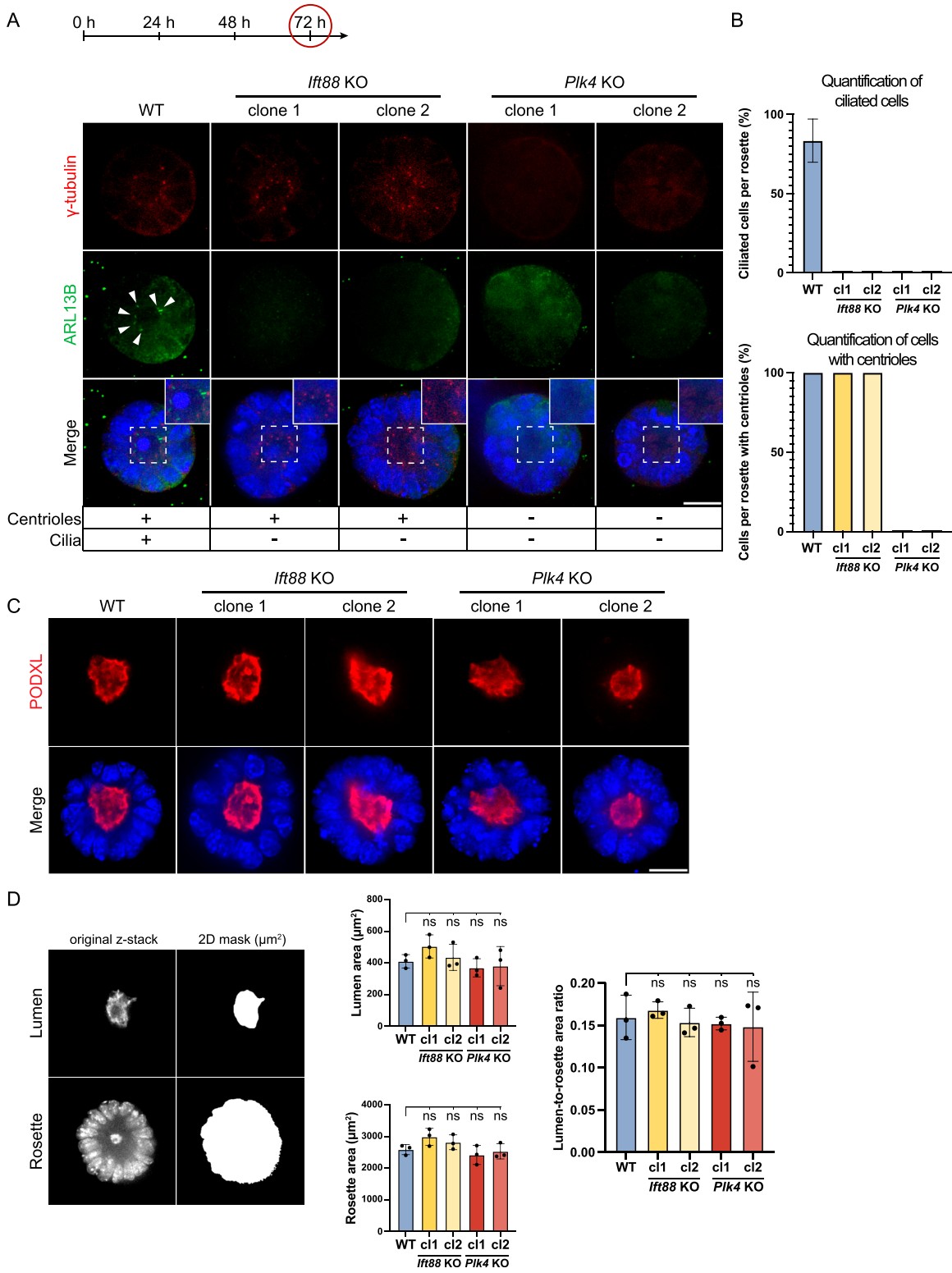

**Figure 4. In *Ift88* and *Plk4* KO 3D rosettes, neither cilia, nor centrioles are required to form a lumen.**
**(A)** Validation of *Ift88* and *Plk4* KO clones in BME-embedded 3D in vitro rosettes growing without 2iLIF for 72 h. Representative immunofluorescence staining of rosettes labeled with antibodies against γ-tubulin (red), ARL13B (green), and DAPI staining (blue), analyzed in >200 cells. The central z-plane is depicted. **(A)** The images of the WT in (A) and the one from the WT in Fig 8E are the same, deriving from the same experimental replicate since both *Ift88* KO and the *Cep83Δexon4* cell line were validated using the same WT control. **(B)** The number of cells exhibiting a cilium per rosette (%) was quantified in both *Ift88* KO and *Plk4* KO rosettes in >100 cells based on ARL13B (green) and DAPI staining (blue). Cells per rosette with centrioles (%) were determined in *Plk4* KO rosettes based on >100 cells based on immunofluorescence staining of γ-tubulin (red), ARL13B (green) and DAPI staining (blue). **(C)** Assessment of lumen formation of 3D BME-embedded in vitro rosettes after 72 h. Data represent three independent experiments (n = 3), each comprising 30 rosettes (one image per rosette) per condition, for a total of 90 images per condition. Representative

leads to the failure of IFT component recruitment and consequently loss of ciliary signaling (Joo et al, 2013). We therefore investigated the role of CEP83 in early mouse embryonic development using two different CRISPR/Cas9-KO strategies (Fig 7B). In KO strategy 1, we designed two gRNAs targeting exon 3 and 4, and in KO strategy 2, we targeted exon 4 along with the following intron (Fig 7B). After genotyping via PCR (Fig S6A), we designed primers to extract the *Cep83* cDNA both from the WT and the different KO clones. The WT showed the expected full-length *Cep83* construct whereas strategy 1 resulted in a frame shift generating multiple stop codons. However, the cDNA extracted from clones generated through KO strategy 2 consistently showed a version of *Cep83* where exon 4 was simply skipped and not included in the cDNA (Fig S6B). We therefore labeled these clones *Cep83Δexon4*. We conducted RT-qPCRs in mESCs to exclude that the KO of *Cep83* induces differentiation (Fig S3A and B). Our data show no significant difference between KO cells and the naive WT control.

We hypothesized that loss of this exon could be sufficient to lead to loss of cilia; therefore, we validated the functional depletion of cilia in these cell lines via immunofluorescence staining with antibodies against γ-tubulin and ARL13B. The cells were starved for 48 h to increase ciliogenesis for imaging experiments. All tested *Cep83$^{-/-}$* and *Cep83Δexon4* clones failed to form cilia, as indicated by the absence of ARL13B expression (Fig 7C and D). Whereas ~5% of WT cells displayed a primary cilium, none of the KO cells analyzed exhibited cilia based on ARL13B staining (Fig 7E). We next evaluated the contribution of CEP83-depleted cell lines to generate polarized, elongated mouse gastruloids. All tested *Cep83$^{-/-}$* and *Cep83Δexon4* clones continued to grow anterior-posterior elongated gastruloids up to 120 h (Fig 7F and G). Furthermore, all *Cep83$^{-/-}$* and *Cep83Δexon4* clones displayed a difference in gastruloid shape between 96 and 120 h comparable to *Ift88* KO clones, most prominently at 96 h, indicated by higher roundness (Fig 7F and G) and a decrease in length (Fig S7A and B). These data suggest that loss of cilia via *Ift88* or *Cep83* KO does not impair gastruloid formation, but delays gastruloid growth and elongation at around 96 h. To investigate whether this is also reflected in the spatiotemporal distribution of germ layer markers, we fixed gastruloids 72, 96, and 120 h after seeding and stained them with antibodies against Brachyury and SOX2 (in three independent experiments, comprising ≥ 20 gastruloids/experiment). *Cep83$^{-/-}$* and *Cep83Δexon4* gastruloids showed a similar spatiotemporal distribution of both markers compared with WT at 72 and 120 h, with less distinct foci at the posterior end at 96 h, similar to the *Ift88* KO (Fig S7C and D). They expressed mesoderm and neural progenitors at the posterior end, indicating anterior-posterior polarization. In summary, our findings demonstrate that depletion of cilia neither through *Ift88* KO nor through the loss of *Cep83* disrupts gastruloid differentiation.

## Naive *Cep83Δexon4* mESCs, initially devoid of cilia, successfully regenerate cilia in rosette and gastruloid differentiation

The comparison of 2D cultures to more sophisticated 3D systems frequently highlights discrepancies between these culture systems with relevance of context-dependent tissue architecture. Fundamental distinctions include extracellular matrix interactions, nutrient availability, and cellular polarization, which are essential features of the 3D microenvironment lacking in 2D cultures (Duval et al, 2017; Kapałczyńska et al, 2018). Since *Cep83Δexon4* clones still express a truncated form of CEP83, we conducted a reassessment of cilium depletion of *Cep83$^{-/-}$* and *Cep83Δexon4* clones in gastruloids. As expected, *Cep83$^{-/-}$* gastruloids did not exhibit cilia after 120 h (Fig 8A). However, *Cep83Δexon4* gastruloids expressed rod-shaped ciliary structures indistinguishable from WT, with a γ-tubulin and ARL13B localization pattern and ciliary length (~2 µm) comparable to the WT (Fig 8B). Time course experiments indicated the gradual initiation of ciliogenesis in *Cep83Δexon4* gastruloids, beginning at 72 h—a time point at which ciliogenesis increased in the WT—albeit at initially lower levels compared with the WT (Fig 8C). However, by 120 h, cilia levels in the mutant reached those observed in the WT. This finding was unexpected, as cilia were consistently absent in undifferentiated *Cep83Δexon4* mESCs in 2D (Fig 7C and D). We investigated whether *Cep83Δexon4* acquires a functional role during 3D gastruloid formation and whether this effect is dependent on its expression levels. To determine whether the lack of cilia in *Cep83Δexon4* mESCs and the delayed initiation of ciliogenesis in *Cep83Δexon4* gastruloids can be rescued by *Cep83Δexon4* overexpression, we designed *Flag-Cep83/Cep83Δexon4* overexpression mESC lines. Immunostaining showed that overexpression of full-length CEP83 in the *Cep83Δexon4* cell line successfully rescued cilium formation (Fig S8A). However, overexpression of CEP83Δexon4 failed to restore ciliogenesis, suggesting that CEP83Δexon4 lacks functional activity in 2D mESCs, even upon overexpression. We next investigated whether CEP83Δexon4 overexpression can rescue the delayed onset of ciliogenesis in gastruloids (Fig S8B). Quantification of cilia in gastruloids fixed at 72 h indicated the most prominent difference in ciliation between WT and *Cep83Δexon4* cell lines. Overexpression of truncated CEP83Δexon4 in the *Cep83Δexon4* cell line partially rescued cilium formation but failed to restore cilium levels similar to the WT. In contrast, overexpression of full-length CEP83 in CEP83Δexon4 gastruloids fully restored cilium levels comparable to WT. These findings suggest that CEP83Δexon4 overexpression can partially rescue ciliation in mouse gastruloids at 72 h, however, not up to WT levels.

To understand why exon 4 of *Cep83* becomes dispensable for ciliogenesis in gastruloids, we investigated the reliance of cilium restoration on exit of naive pluripotency by differentiating mESCs into EpiLCs in 2D culture for 48 h. Cilia were absent in these differentiated *Cep83Δexon4* clones, suggesting that this phenomenon

---

immunofluorescence staining labeled with antibodies against PODXL (red) and DAPI staining (blue). The central z-plane is depicted. **(D)** Quantification of lumen and rosette size based on the following masking model: 2D mask (area) of the central z-plane, using PODXL as a lumen marker and DAPI to assess rosette size. Column charts show lumen area, rosette area, and lumen-to-rosette area ratio of the central lumen and rosette z-planes. Data represent three independent experiments (n = 3), each comprising 30 rosettes (one image per rosette) per condition, for a total of 90 images per condition. (ns, non-significant, $^{ns}P > 0.05$, unpaired *t* test). Scale bar (A, C): 20 µm.

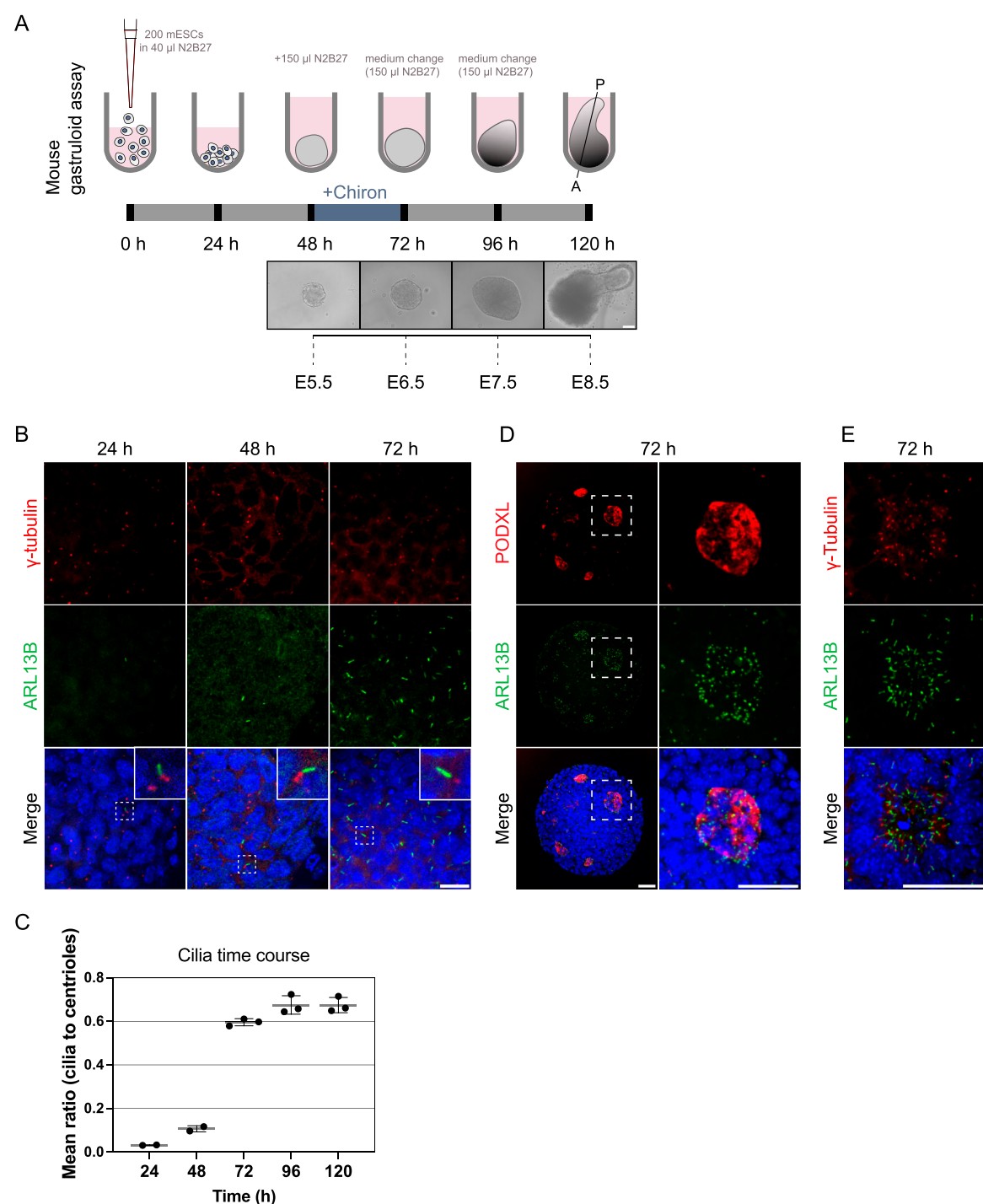

**Figure 5. Gastruloids recapitulate early and late ciliogenesis in vitro.**
**(A)** Mouse gastruloid protocol up to 120 h of cultivation with representative brightfield images of different time points, compared with corresponding stages of in vivo mouse embryonic development. Mouse embryonic stem cells are cultivated in u-bottom, ultra-low attachment plates for 48 h in N2B27, followed by a Chiron pulse for 24 h, leading to symmetry break, anterior-posterior polarization and generation of all three germ layers, based on Beccari et al (2018). **(B)** Data represent two independent experiments (24 and 48 h) and three independent experiments (72, 96, and 120 h) of at least 21 images per experiment (three images per gastruloid, including the top section, middle, and bottom of the gastruloid and anterior/posterior sections). Representative example of immunofluorescence staining of gastruloids at 24, 48, and 72 h, labeled with antibodies against γ-tubulin (red), ARL13B (green), and DAPI staining (blue). **(B)** The images of the WT in (B) and the ones from the WT in Fig 8C are the same, deriving from the same experimental replicate since both figures derive from the same experimental replicate. **(B, C)** Quantification of data from (B): the mean ratio (cilia to centrioles) over a time course of 24–120 h, showing the increase of ciliogenesis during gastruloid development. Dots represent the mean ratio of individual experimental replicates per time point (including at least 30 gastruloids per time point), with standard deviations indicated. **(D)** Lumenogenesis and ciliogenesis in gastruloids. Data represent three independent experiments (72 h) of at least 21 gastruloids (three images per gastruloid). Representative example of immunofluorescence staining of gastruloids at 72 h, labeled with antibodies against PODXL, ARL13B (green), and DAPI staining (blue). Maximum intensity projection of central z-planes. **(E)** Ciliogenesis in gastruloids. Data represent three independent experiments (72 h) of at least 21 gastruloids (three images per gastruloid).

is not driven by the transition out of naive pluripotency, but rather by an alternative mechanism (Fig 8D). We hypothesized that cell polarization, establishment of the extracellular matrix and morphogens, present in a 3D environment, enable ciliogenesis independent of exon 4 of *Cep83*. To test this, we differentiated *Cep83Δexon4* cells into 3D in vitro rosettes, which polarize and form lumen during exit of naive pluripotency but without differentiating further into all germ layers. 3D rosettes, derived from *Cep83Δexon4* clones, expressed cilia after 72 h (Fig 8E). These data suggest that the truncated isoform of CEP83 is non-functional under 2D conditions. In contrast, *Cep83Δexon4* cells grown in 3D rosettes or gastruloids form cilia, indicating that the CEP83Δexon4 variant is able to bypass potential limitations encountered in 2D. However, the mechanism enabling CEP83Δexon4 activity in differentiation remains unclear and requires further investigation in the future.

In summary, we use different 3D model systems that serve as novel tools to study ciliogenesis in early embryonic development, from implantation to gastrulation, recapitulating not only key developmental events but also context-specific cilium assembly. Our study highlights the importance of incorporating complex 3D environment setups in developmental studies and their advantages over less advanced 2D systems.

## Discussion

Primary cilia and centrioles play essential roles in mammalian embryonic development, yet their precise contributions to embryonic signaling and morphology in the early embryo before the formation of the node remain incompletely understood. In mice, centrioles are initially absent in the fertilized embryo and first assemble de novo at the blastocyst stage at E3.5 (Gueth-Hallonet et al, 1993; Courtois et al, 2012; Howe & FitzHarris, 2013). Yet, only a short time later they are clearly functionally important, with their loss resulting in embryonic arrest by E7.5 (Hudson et al, 2001). Meanwhile, primary cilia first arise on epiblast cells during cavitation at E5.5-E6 (Bangs et al, 2015), and ciliary mutants exhibit mid-gestation arrest (E11) because of defects in Hh-dependent neural and limb patterning (Huangfu et al, 2003; Cortellino et al, 2009). It is essential to understand the processes occurring between the initiation of centriole and cilium formation and the onset of embryonic lethality to elucidate their developmental roles. In this study, we used different 3D in vitro model systems to study centrioles and cilia in early mouse embryonic development and their contribution to polarization, germ layer differentiation, and symmetry breaking.

Recent advances in cilium research highlight the close interplay between cell differentiation and ciliogenesis (Yanardag & Pugacheva, 2021), with primary cilia acting as critical signal transducers that regulate cell differentiation in various tissues (Arrighi et al, 2017; Shim et al, 2023; Coschiera et al, 2024). We investigated whether cell differentiation per se can promote cilium formation. The widely used transition from mESCs to EpiLCs reflects many transcriptional changes observed in the early mouse embryo; however, this transition did not enhance ciliogenesis, suggesting that increased primary cilium formation is not an inherent feature of exit from naive pluripotency.

In the 3D rosette model system, where the differentiating mESCs are embedded into extracellular matrix, cells similarly exit from naive pluripotency, whereas also undergoing morphological changes including apico-basal polarization and lumenogenesis, recapitulating aspects of the formation of the proamniotic cavity in the embryo (Bedzhov & Zernicka-Goetz, 2014; Shahbazi et al, 2017; Kim et al, 2021). Under these conditions, the differentiating mESCs exhibit increased ciliation, demonstrating further that ciliation is not coupled to differentiation per se but potentially to the spatial and mechanical cues provided by the 3D microenvironment.

We generated centriole (*Plk4*) and cilium (*Ift88*) KO cell lines, to study the function of centrioles and cilia in the 3D in vitro rosette model system. Loss of *Plk4* and *Ift88* did not disrupt polarization and lumenogenesis, with the establishment of well-organized rosettes of similar size after 72 h of 3D rosette culture. Our results demonstrate that centrioles and cilia are not required for 3D in vitro differentiation and lumenogenesis during early mouse development, consistent with the reported embryonic arrest of *Plk4*$^{-/-}$ mice only after gastrulation (Hudson et al, 2001; Swallow et al, 2005). The purpose of cilium formation during early developmental stages remains unclear, as cilia do not appear to contribute to the establishment of rosette morphology. However, evidence from human cells shows that ablation of primary cilia via Tau tubulin kinase 2 (TTBK2) depletion does not impact the maintenance of undifferentiated human pluripotent stem cells (hPSCs) but does affect rosette size during neural differentiation, suggesting a "poised state" of the ciliated cells in the undifferentiated state to quickly respond to cues enabling a specific differentiation program (Binó & Čajánek, 2023). Furthermore, it has been shown that KIF3A$^{-/-}$ and KIF3B$^{-/-}$ hPSCs lacking cilia remain pluripotent and self-renewing, but during differentiation they develop defects in neurogenesis, nephrogenesis, and kidney cyst formation. These abnormalities appear only after prolonged differentiation in complex tissues and organoids, suggesting cilia are crucial for organizing higher order tissue architecture, likely by coordinating cell migration (Cruz et al, 2022).

Whereas 3D rosettes are a great model system to study cell state changes during the implantation stage, cells will not undergo further developmental changes normally observed during gastrulation. Therefore, we used mouse gastruloids, a powerful model system recapitulating germ layer specification and in vivo development up to E8.5 (Beccari et al, 2018; Hashmi et al, 2022; Stelloo et al, 2024). After the 72 h Chiron pulse, we noted a marked increase in ciliation. In vivo, ciliated epithelial cells typically orient their cilia toward a lumen (van der Vaart et al, 2021) and in gastruloids, we observed PODXL-positive cavities with apically enriched primary cilia projecting into these compartments. Although gastruloids lack extraembryonic tissues and do not form a central, pro-

---

Representative example of immunofluorescence staining of gastruloids at 72 h, labeled with antibodies against γ-tubulin, ARL13B (green), and DAPI staining (blue). Maximum intensity projection of central z-planes. Scale bar (A): 100 $\mu m$. Scale bar (B): 20 $\mu m$. Scale bar (D, E): 40 $\mu m$.

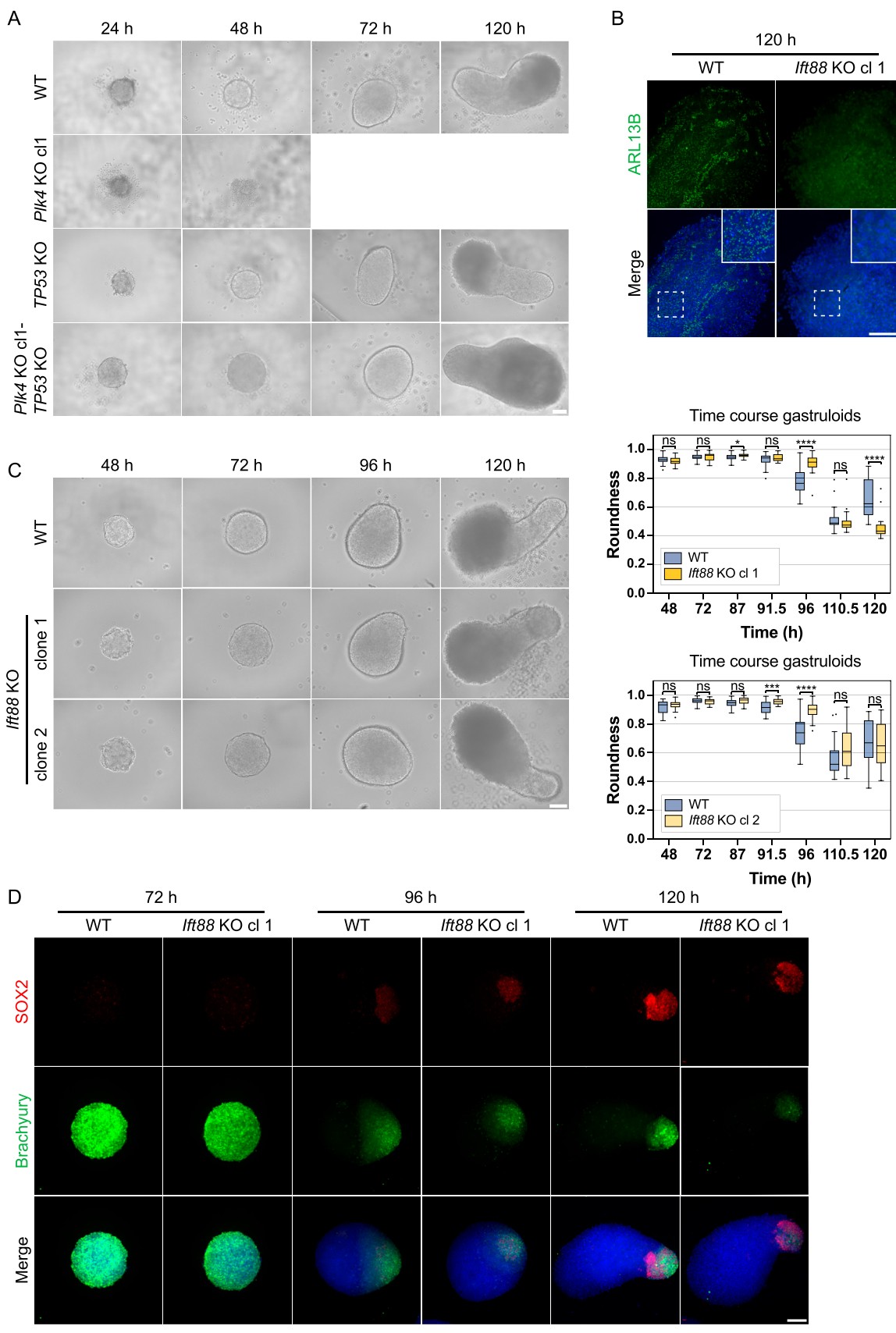

amniotic cavity (Beccari et al, 2018; Hashmi et al, 2022; Stelloo et al, 2024), these luminal spaces likely correspond to localized, tissue-specific epithelial cavities similar to transient lumens seen during organogenesis in vivo (Datta et al, 2011; Kim & Bedzhov, 2022). The apical positioning of cilia suggests these structures may serve as signaling hubs within epithelialized domains, potentially contributing to regional patterning despite the absence of full embryonic architecture.

*Plk4* KO mouse gastruloids disassembled from the time of seeding onward, before the Chiron pulse, and failed to progress further in development. Centrosomal activity is essential for cell cycle progression (Hinchcliffe et al, 2001; Khodjakov & Rieder, 2001). Several studies have shown that loss of centrioles results in cell cycle arrest and p53 dependent apoptosis (Bazzi & Anderson, 2014; Lambrus et al, 2015; Wong et al, 2015; Grzonka & Bazzi, 2024). To assess the p53 dependence of the *Plk4* KO gastruloid phenotype, we generated a *Plk4* KO-*TP53* KO double-KO cell line. Notably, the double KO exhibited a morphological rescue, producing elongated gastruloids comparable to those derived from WT and *TP53* KO cells for up to 120 h. These findings indicate that the pronounced gastruloid defects observed in the *Plk4* KO are p53-dependent, consistent with observations reported for Sas-4 (Xiao et al, 2020 *Preprint*) and Sas-6 (Grzonka & Bazzi, 2024). We hypothesize that similar to Sas-4$^{-/-}$ mouse embryos (Xiao et al, 2020 *Preprint*), centriole loss through *Plk4* KO might activate a p53-dependent mitotic surveillance pathway in gastruloids, a possibility that warrants further study.

One proposed mechanism causing this arrest is the increased duration of mitosis in centriole-depleted mutant mouse embryos (Bazzi & Anderson, 2014). The duration of mitosis is monitored during cell cycle progression; extended mitosis, including cumulatively over multiple cell cycles, has been associated with G1 arrest, an ability that is often lost in p53-mutant aberrant cells (Meitinger et al, 2024). In hPSCs, inactivation of the PLK4-STIL module causes progressive centrosome loss, resulting in prolonged, error-prone acentrosomal mitosis and p53 stabilization. Whereas p53 up-regulation does not trigger significant apoptosis, it promotes loss of self-renewal and induction of differentiation (Renzova et al, 2018).

Interestingly, in *Drosophila melanogaster*, a delay in mitotic spindle assembly following loss of centrioles has also been observed, with the key difference that these embryos develop to the adult stage without reported cell death and only die postnatally due to the absence of cilia (Basto et al, 2006). This suggests that vertebrates possess more stringent cell cycle checkpoints to eliminate acentriolar cells.

Although the observed disassembly in our *Plk4* KO gastruloids resembles embryonic arrest of in vivo data at E7.5 (Hudson et al, 2001), the onset of the phenotype occurs earlier than expected, with gastruloids gradually disassembling from 48 h on, roughly corresponding to E5.5-E6.5. The discrepancy between the timing of the phenotype observed in vitro and in vivo could be attributed to the gastruloid model system itself, which is derived from mESCs that possess fully functional and matured centrioles, whereas these are typically absent until the blastocyst stage in vivo (Gueth-Hallonet et al, 1993; Courtois et al, 2012; Howe & FitzHarris, 2013). This difference may account for the accelerated onset of the phenotype in our model. In addition to its role in centriole duplication, PLK4 has been reported to have a centriole-independent function in early mouse embryos, where it promotes acentriolar spindle assembly in mammalian oocytes (Coelho et al, 2013; Bury et al, 2017). However, our data show that the loss of *Plk4* has no effect on cell viability in mESCs.

Whereas centriole depletion significantly impaired gastruloid formation, cilia loss in *Ift88* KO mESCs did not perturb the formation of elongated anterior-posterior differentiated gastruloids, exhibiting a spatiotemporal expression of mesoderm (Brachyury) and neuronal progenitor cells (SOX2) similar to the WT. It has been previously demonstrated that cilium KO mouse embryos undergo arrest at mid-gestation around E11 (Huangfu et al, 2003; Cortellino et al, 2009). The absence of a severe phenotype in *Ift88* KO gastruloids may be attributed to the fact that they can only be cultured up to 120 h without shaking, thereby recapitulating developmental stages only from E5.5 to E8.5 as previously described (Beccari et al, 2018; Arias et al, 2022; Stelloo et al, 2024).

Interestingly, although *Ift88* KO gastruloids were able to elongate up to 120 h, they exhibited a difference in morphology at 96 h, appearing rounder and less elongated compared with controls. This effect was consistently observed across all tested clones that lacked cilia. In WT gastruloids, only a small fraction of cells is ciliated up to 48 h. The number increases six-fold after 72 h, corresponding to the time frame cilia first appear in the mouse embryo at E5.5–E6.5 (Bangs et al, 2015). After 72 h, ciliogenesis reached a plateau and remained stable throughout the culture period until 120 h. The timing of the potential delay in gastruloid elongation coincides with the peak of ciliogenesis in WT gastruloids and may be attributed to the absence of ciliary signaling. Hh signaling, for example, has been shown to be impaired in cells

---

**Figure 6.  Gastruloids continue to elongate without cilia in the *Ift88* KO but disassemble in the centriole-depleted *Plk4* KO model.**
**(A)** Representative brightfield images of WT, *TP53* KO, *Plk4* KO, and *Plk4* KO-*TP53* KO gastruloids 24, 48, 72, and 120 h after seeding. Data represent two independent experiments (n = 2), comprising of a total of 72 gastruloids per condition. In addition, for WT and *Plk4* KO gastruloids, additional gastruloid data were acquired, comprising three additional independent experiments (n = 3), each consisting of 24 gastruloids per condition/experiment. **(A)** The brightfield images of WT and *TP53* KO in (A) are the same as in Fig S5B, deriving from the same experimental replicate. **(B)** Immunofluorescence staining of WT and *Ift88* KO gastruloids at 120 h, labeled with antibodies against ARL13B (green) and DAPI staining (blue), measured in three independent experiments. The maximum intensity projection is depicted. **(C)** Representative brightfield images of WT and *Ift88* KO gastruloids 48–120 h after seeding. Data represent three independent experiments (n = 3), each comprising 24 gastruloids per condition, for a total of 72 gastruloids per condition. Box plots indicating the roundness of WT and *Ift88* KO gastruloids (with one representative experiment depicted). The box represents the interquartile range (IQR), with the median indicated by a horizontal line. Whiskers extend to 1.5 × IQR, and data points beyond this range are considered outliers, shown as dots ($^{ns}P > 0.05$, $****P < 0.0001$, $***P < 0.001$, unpaired *t* test). **(D)** Immunofluorescence staining of WT and *Ift88* KO gastruloids at 72, 96, and 120 h, labeled with antibodies against SOX2 (red), Brachyury (green), and DAPI staining (blue). The maximum intensity projection is depicted. Data represent three independent experiments (n = 3), each comprising at least 17 gastruloids (one image per gastruloid) per time point and cell line. Scale bar (A, B, C, D): 100 $\mu m$. WT, parental cell line.

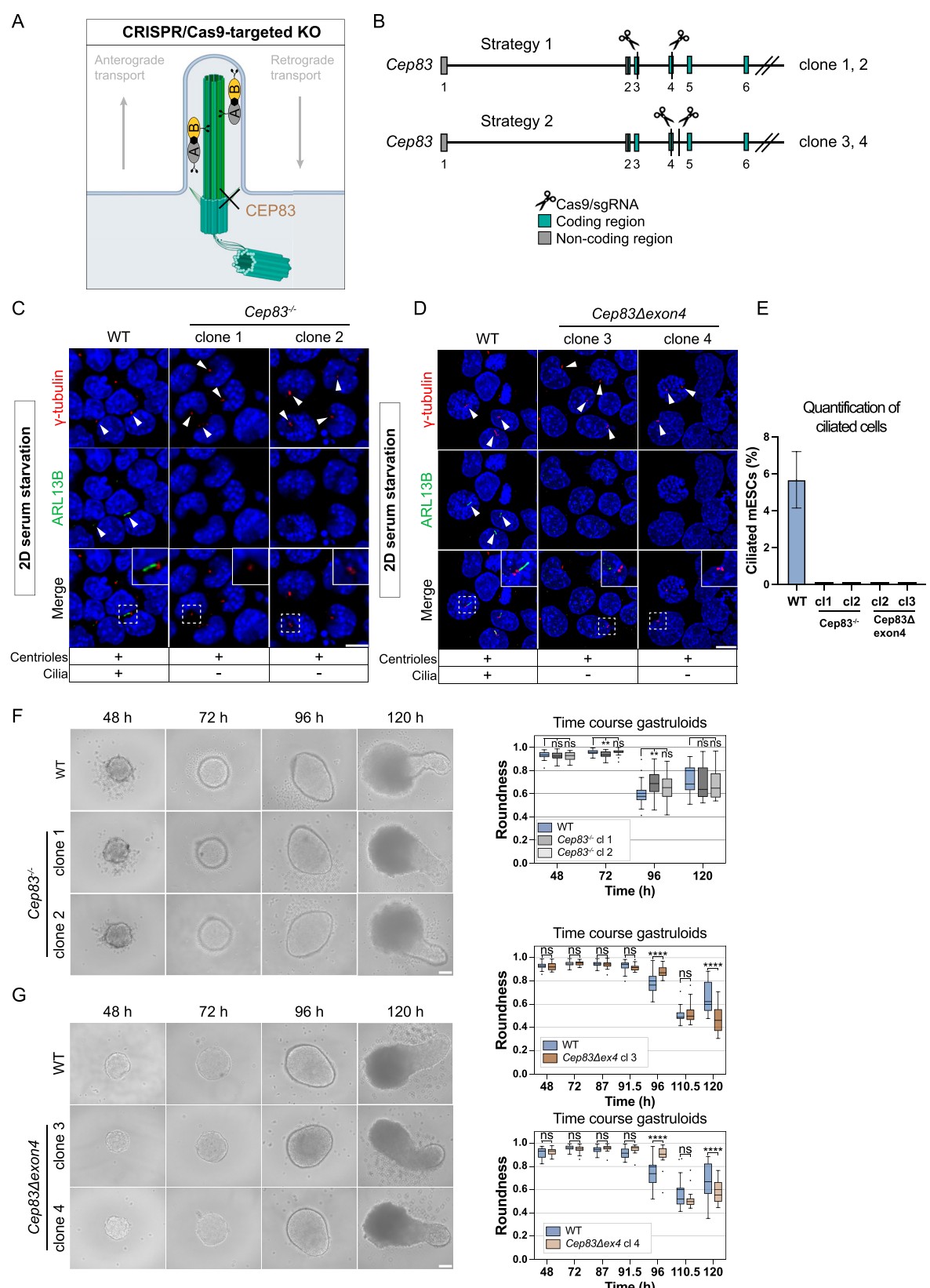

**Figure 7. _Cep83_ is not essential for gastruloid formation.**
**(A)** Scheme of a primary cilium indicating the CRISPR/Cas9 KO target _Cep83_. CEP83 is a distal appendage protein (DAP), required for anchoring the mother centriole to the plasma membrane and hence for ciliogenesis. **(B)** Strategy for the _Cep83_ KO generation in Mouse embryonic stem cells. gRNA pairs are targeting exon 3 and 4, and exon 4 and the following intron. Coding regions (green) and non-coding regions (gray) are indicated. **(C, D)** IF-validation of cilium depletion in _Cep83−/−_ and _Cep83Δexon4_

with disrupted IFT, leading to abnormal limb and neural tube patterning (Huangfu et al, 2003; Haycraft et al, 2005; Huangfu & Anderson, 2005; Liu et al, 2005). In addition, Wnt signaling is involved in different aspects of development, including establishment of an anterior-posterior axis, cell fate decisions, proliferation, and cell death (Vuong & Mlodzik, 2023). Whether the loss of ciliary signaling pathways in *Ift88* KO gastruloids underlies the observed morphological changes and why this phenotype does not persist up to 120 h remains to be determined. Our gastruloid cultivation is limited to 120 h, after which, and occasionally even earlier, disassembly begins. The loss of phenotype might therefore reflect a general loss of gastruloid integrity rather than the existence of compensatory mechanisms. Distinguishing between these possibilities would require conditions allowing gastruloids to be cultured beyond 120 h using alternative long-term cultivation protocols.

In addition to our studies with *Plk4* and *Ift88* KO cell lines in different in vitro model systems, we investigated the role of the centriolar distal appendage protein CEP83 in early mouse development. Distal appendages are involved in anchoring the mother centriole to the cell membrane (Tanos et al, 2013; Lo et al, 2019; Chong et al, 2020), and *Cep83* KO completely inhibits ciliogenesis, preventing IFT recruitment and resulting in a loss of ciliary signaling (Joo et al, 2013). As anticipated, our data demonstrate that *Cep83*$^{-/-}$ cell lines are capable of generating polarized, elongated mouse gastruloids, similar to what was observed in *Ift88* KO clones. Limited evidence indicates that ciliogenesis might still take place in CEP83 mutants, although it appears to be less efficient and is frequently associated with structural defects (Shao et al, 2020; Mansour et al, 2022). In our study, *Cep83*$^{-/-}$ cells lacked cilia, whereas the truncated isoform *CEP83Δexon4*, unexpectedly recovered ciliogenesis during 3D differentiation in gastruloids. Ciliation occurred gradually in gastruloids and reached WT levels at 120 h. This suggests that exon 4 of CEP83 is important for ciliogenesis in non-polarized cells, such as during 2D cultivation of mESCs, but is only partially required during 3D differentiation. Whereas overexpression of the *Cep83Δexon4* variant did not rescue ciliogenesis in 2D, overexpression of the isoform in gastruloids partially rescued ciliation, revealing a context-dependent function of exon 4. Key differences between 2D and 3D culture systems include cell polarization, alterations in gene expression and signaling, extracellular matrix (ECM) composition, mechanical forces, nutrient and oxygen gradients, and the nature of cell-cell and cell-ECM interactions. These factors may influence the ability of truncated CEP83 at the basal body to anchor effectively to the

cell membrane. To disentangle some of these potential contributors, extensive analysis of distal appendage protein localization under 2D and 3D conditions, along with further assessment of ciliogenesis on both polarizing and nonpolarizing substrates, will be necessary in the future.

Regardless of the precise mechanism, this discrepancy highlights the importance of studying cellular processes in their proper 3D context, with conventional 2D cultures potentially yielding misleading results. Whereas 3D rosettes are a powerful model system to study cell state changes during the implantation stage (Bedzhov & Zernicka-Goetz, 2014; Shahbazi et al, 2017; Kim et al, 2021) and even recapitulate early events of primitive streak differentiation (Sato et al, 2024), gastruloids mimic germ layer specification and in vivo development up to E8.5 (Beccari et al, 2018; Hashmi et al, 2022; Stelloo et al, 2024). However, being generated from stable cell lines and lacking extraembryonic tissues, they do not precisely mirror the in vivo timing of ciliogenesis but recapitulate early events of ciliogenesis in a controlled, embryonic tissue context. For a more comprehensive investigation of early ciliogenesis that includes contributions from extraembryonic lineages, in vitro ETS (Harrison et al, 2017) or ETX (Dupont et al, 2023) models may provide a more physiologically representative platform.

In conclusion, our study presents the first in vitro study of centriole and cilium formation in early mouse development, employing different 3D models to recapitulate key events of implantation, tissue patterning, and anterior-posterior elongation. Although these models do not represent true embryos, they provide valuable platforms for investigating centriole and cilium formation and function within a highly controlled and closely monitored experimental framework.

# Materials and Methods

Details of reagents, cell lines, equipment, software, recombinant DNA, gRNA sequences, primers, RT-qPCR primers, and antibodies are listed in the file lists—Methods (Table S1).

### Cell culture maintenance

Male R1 mESCs (Nagy et al, 1993, kindly shared by Wysocka lab, Stanford) were routinely cultivated at 37°C, 5% $CO_2$ in "2iLIF medium," composed of N2B27 base medium—HyClone DMEM/F12 1:1 mix (SH30271.FS; Cytiva) with 2.5 mM L-Glutamine and without

---

clones. Representative example of immunofluorescence staining of WT, Cep83$^{-/-}$, and *Cep83Δexon4* clones after induced ciliogenesis (48 h of starvation), labeled with antibodies against γ-tubulin (red), ARL13B (green), and DAPI staining (blue). Maximum intensity projection of central z-planes. **(D)** The images of the WT in (D) and the one from WT in Fig 3C are the same, deriving from the same experimental replicate since both *Ift88* KO and the *Cep83Δexon4* cell line were validated using the same WT control. **(E)** Ciliated cells (%) are indicated. At least 100 cells (n ≥ 100) were quantified based on immunofluorescence staining of γ-tubulin (red), ARL13B (green), and DAPI staining (blue). **(F)** Representative brightfield images of WT and *Cep83*$^{-/-}$ gastruloids 48–120 h after seeding. Box plot showing the roundness of WT and *Cep83*$^{-/-}$ gastruloids. The box represents the interquartile range (IQR), with the median indicated by a horizontal line. Whiskers extend to 1.5 × IQR, and data points beyond this range are considered outliers (shown as dots). n = 3 independent experiments with one representative experiment depicted ($^{ns}P > 0.05$, ****$P < 0.0001$, **$P < 0.01$, unpaired *t* test). **(G)** Representative brightfield images of WT and *Cep83Δexon4* gastruloids 48–120 h after seeding. Box plot showing the roundness of WT and *Cep83Δexon4* gastruloids. The box represents the interquartile range (IQR), with the median indicated by a horizontal line. Whiskers extend to 1.5 × IQR, and data points beyond this range are considered outliers (shown as dots). n = 3 independent experiments with one representative experiment depicted. Scale bar (C, D): 10 $\mu$m. Scale bar (F, G): 100 $\mu$m. WT, parental cell line.

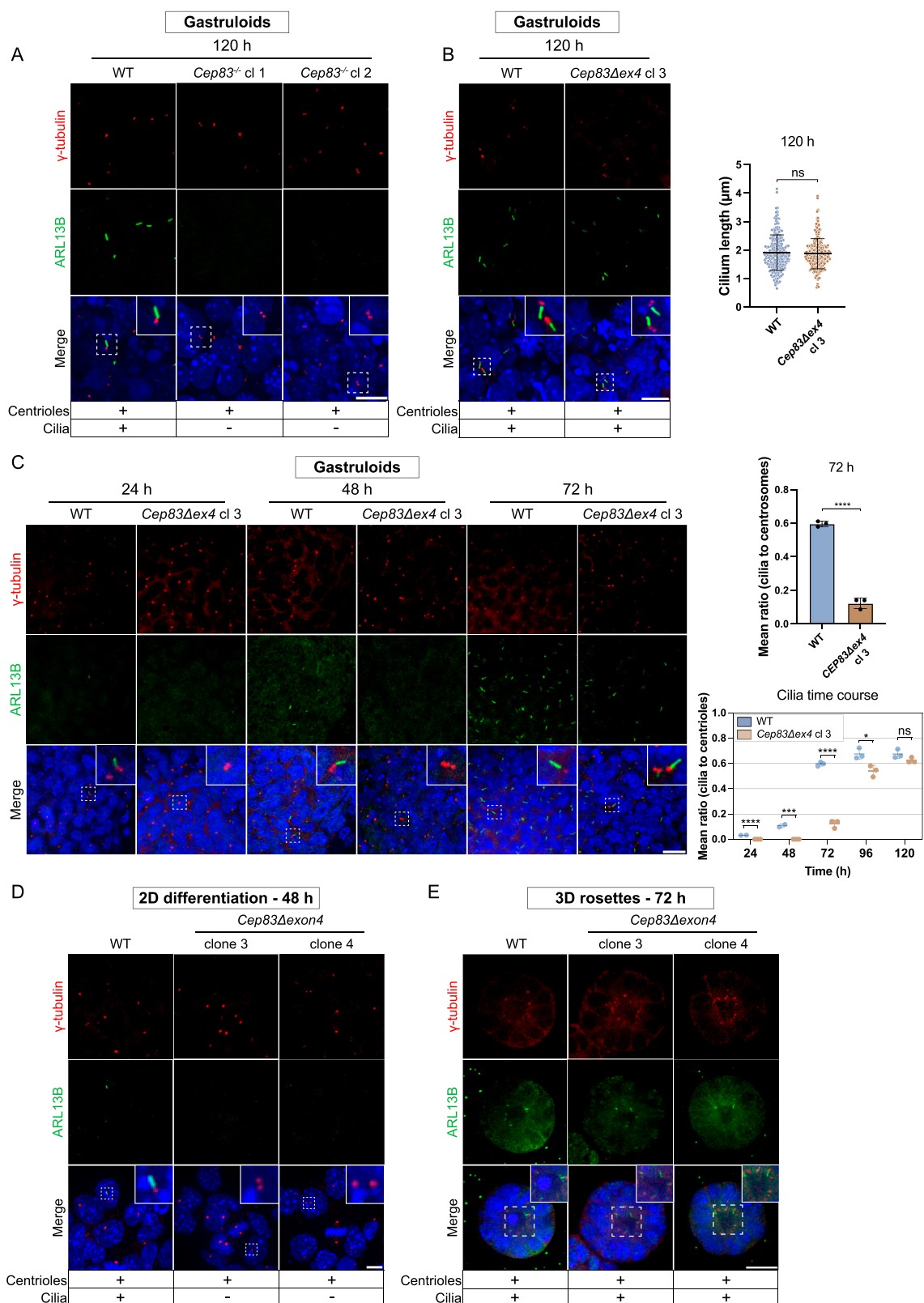

**Figure 8. Cilium-depleted, naive *Cep83Δexon4* Mouse embryonic stem cells regain cilia in 3D rosettes and gastruloids.**
**(A)** Immunofluorescence staining of WT and *Cep83⁻/⁻* gastruloids at 120 h, labeled with antibodies against γ-tubulin (red), ARL13B (green), and DAPI staining (blue). The maximum intensity projection of the five central z-planes is depicted, analyzed in at least 24 gastruloids. **(B)** Immunofluorescence staining of WT and *Cep83Δexon4* gastruloids at 120 h, labeled with antibodies against γ-tubulin (red), ARL13B (green), and DAPI staining (blue). Data were analyzed in at least 24 gastruloids. The central z-plane is depicted. Cilium length (μm) was quantified in at least 229 cilia per condition (mean + SD, ns, non-significant, $^{ns}P > 0.05$, unpaired t test). **(C)** Representative

Hepes, 4 g/liter AlbuMAXTM II (11021029; Gibco), 1x serum-free B-27 Supplement (17504044; Gibco), 1x N2 components (home-made, Sigma-Aldrich, R&D Systems), 1x MEM NEAA (11140035; Gibco), 1x Penicillin-Streptomycin (15070063; Gibco), 1x Sodium Pyruvate (11360039; Gibco), 0.055 mM 2-Mercaptoethanol (21985023; Gibco), supplemented with the MEK inhibitor Mirdametinib (PD0325901, 0.8 $\mu$M; MedChemExpress), the GSK3$\beta$ inhibitor Laduviglusib (CHIR99021, 3.3 $\mu$M; MedChemExpress) and 10 ng/ml human LIF (provided by the VBCF Protein Technologies Facility, https://www.viennabiocenter.org/facilities/) on CytoOne Multi-Well Plates. Before usage, tissue culture dishes were pre-coated with 7.5 $\mu$g/ml Poly-L-ornithine hydrobromide (P4638; Sigma-Aldrich) for 1 h at 37°C, followed by 5 $\mu$g/ml Laminin (L2020; Sigma-Aldrich) for 1 h at 37°C. The cells were passaged every 2 to 3 d in an appropriate ratio using 1x Trypsin-EDTA solution (T3924; Sigma-Aldrich) for 3 min at 37°C to detach the cells and 10% FBS (F7524; Sigma-Aldrich) to stop the reaction. Cells were regularly tested for mycoplasma contamination using the MycoAlertTM PLUS Mycoplasma Detection Kit (LT07-705; Lonza).

### mESC to EpiLC differentiation (2D)

For differentiation of naive mESCs to formative EpiLCs, 10,000 cells/well were seeded in N2B27 without 2iLIF on $\mu$-Slide eight Well Glass Bottom Plates (80827; ibidi) coated with fibronectin (#663, 10 $\mu$g/ml; YO Proteins). Undifferentiated controls were seeded in parallel and cultured in 2iLIF. After 48 h of growth, cells were fixed and stained for immunofluorescence microscopy according to the protocol below (fixation and immunofluorescence staining of 2D cultured cells).

### Cell starvation (2D)

10,000 cells were seeded in 2iLIF on fibronectin-coated (10 $\mu$g/ml) $\mu$-Slide 8 Well Glass Bottom Plates (80827; ibidi). After 24 h, cells were washed once with 250 $\mu$l Dulbecco's Phosphate Buffered Saline (PBS, D8537; Sigma-Aldrich), and the starvation was initiated by addition of 250 $\mu$l starvation medium (DMEM/F12, supplemented with 0.1% FBS, 0.8 $\mu$M MEK inhibitor PD0325901 and 3.3 $\mu$M GSK3$\beta$ inhibitor CHIR99021). Cells were starved for 24–48 h, as indicated in the figure legends. The controls remained in 2iLIF for the respective time. Samples were fixed and stained for immunofluorescence microscopy according to the protocol below (fixation and immunofluorescence staining of 2D cultured cells).

### Fixation and immunofluorescence staining of 2D cultured cells

2D cultured cells were 1x washed with PBS and fixed with 250 $\mu$l PFA (pre-warmed 37°C) for 15 min at RT. After 3x washing with PBST (PBS + 0.1% TWEEN 20), cells were permeabilized with 250 $\mu$l 0.1% Triton-X/PBS for 10 min at RT, washed 3x with PBST and incubated for 30 min in 250 $\mu$l blocking buffer (PBST + 5% BSA) at RT. Next, cells were stained with primary antibody (diluted in blocking buffer) overnight at 4°C. After 3x washing with PBST, cells were incubated with secondary antibody solution for 1 h at RT (negative control received only the secondary antibody). Finally, samples were incubated in DAPI solution (1 $\mu$g/ml, D9542; Sigma-Aldrich) for 10 min, washed 3x in PBS and stored at 4°C until image acquisition.

### 3D in vitro rosette assay

Previously described 3D rosette formation assay (Bedzhov & Zernicka-Goetz, 2014; Shahbazi et al, 2017) was adjusted and optimized for our R1 mESC line to stably recapitulate polarization, lumenogenesis, and cilium formation during mouse implantation. mESCs were cultivated in 2iLIF at 37°C, 5% CO$_2$ and split at least 1x before the rosette formation assay to adjust to our culture conditions. Cells were trypsinized, washed two times in PBS, pelleted (20,000 cells/pellet) and resuspended in 20 $\mu$l ice-cold Cultrex Reduced Growth Factor Basement Membrane Extract (BME), Type 2 (3533-005-02; R&D Systems), which mimics the basement membrane that surrounds the epiblast during implantation (Bedzhov & Zernicka-Goetz, 2014). Each 20 $\mu$l cell suspension drop was carefully placed in the center of a $\mu$-Slide 8 Well Glass Bottom Plate (80827; ibidi) and incubated for 10 min at 37°C to enable the BME to solidify. 250 $\mu$l of pre-warmed N2B27 base medium or 2iLIF medium (depending on the assay) was added to each well and plates cultivated at 37°C in 5% CO$_2$ for up to 72 h with daily medium changes.

### Fixation and immunostaining of rosettes

3D rosettes were washed with 1x PBS and fixed with 250 $\mu$l (pre-warmed 37°C) 4% PFA (methanol free, 0219998380; MP Biomedicals) for 30 min at RT. Cells were washed 3x with PBST (0.1% TWEEN in PBS) and permeabilized with 250 $\mu$l 0.3% Triton-X/PBS for 30 min at RT. After washing 3x with PBST, 250 $\mu$l blocking buffer was added to each well and incubated for 30 min at RT. Next, samples were stained with primary antibodies (diluted in blocking buffer) against Podocalyxin (MAB1556; R&D Systems) for lumen quantifications, ARL13B (17711-1-AP; Proteintech) and $\gamma$-tubulin (T6557; Sigma-

immunofluorescence staining of WT and *Cep83Δexon4* gastruloids at 24, 48, and 72 h, labeled with antibodies against $\gamma$-tubulin (red), ARL13B (green), and DAPI staining (blue). Quantification time course of the mean ratio of cilia to centrioles from 24 to 120 h. Data represent two independent experiments (24 and 48 h) and three independent experiments (72, 96, and 120 h) of at least 21 images per experiment (three images per gastruloid, including the top section, middle, and bottom of the gastruloid and anterior/posterior sections). Dots represent the mean ratio of individual experimental replicates per time point, with standard deviations indicated (ns > 0.05, *P < 0.05, ***P < 0.001, ****P < 0.0001, unpaired *t* test). **(C)** The images of the WT in (C) and the ones from the WT in Fig 5B are the same, deriving from the same experimental replicate since both figures derive from the same experimental replicate. **(D)** Immunofluorescence staining of formative EpiLCs after 48 h of differentiation labeled with antibodies against $\gamma$-tubulin (red), ARL13B (green), and DAPI staining (blue), analyzed in >100 cells. Maximum intensity projection is depicted. **(E)** Validation of *Cep83Δexon4* clones in BME-embedded 3D in vitro rosettes growing without 2iLIF for 72 h. Representative immunofluorescence staining labeled with antibodies against $\gamma$-tubulin (red), ARL13B (green), and DAPI staining (blue), analyzed in >200 cells. The central z-plane is depicted. **(E)** The images of the WT in (E) and the one from the WT in Fig 4A are the same, deriving from the same experimental replicate since both *Ift88* KO and the *Cep83Δexon4* cell line were validated using the same WT control. Scale bar (A, B, D): 10 $\mu$m. Scale bar (C, E): 20 $\mu$m. WT, parental cell line.

Aldrich) for KO validations, at 4°C overnight. After 3x washing with PBST, the secondary antibody Anti-Rat IgG H&L Alexa Fluor 555 (ab150154; Abcam) for lumen quantifications, Anti-Rabbit Alexa Fluor 488 (ab150073; Abcam), and donkey Anti-Mouse Alexa Fluor 555 (ab150106; Abcam) for KO validations, incubated for 1 h at RT, including the negative control, containing only the secondary antibody. Finally, samples were incubated in DAPI solution (1 µg/ml) for 15 min, washed 3x in PBS and stored at 4°C until image acquisition.

## Gastruloid culture

mESCs were cultivated up to 80% confluency in 2iLIF at 37°C, 5% $CO_2$ and passaged 2x before starting the gastruloid assay to provide stable cell growth and ensure cell adaptation to medium conditions. The gastruloid differentiation protocol was previously described (Beccari et al, 2018). mESCs were detached with trypsin, washed 3x with PBS and resuspended in gastruloid N2B27 medium—50% DMEM/F12 with GlutaMAX (31331028; Gibco), 50% Neurobasal medium (21103049; Gibco), 1x GlutaMAX (3133102; Gibco), 1x MEM NEAA (11140035; Gibco), 1x Penicillin-Streptomycin (15070063; Gibco), 1x Sodium Pyruvate (11360039; Gibco), 0.1 mM 2-Mercaptoethanol (21985023; Gibco), 1x N-2 Supplement (17502048; Gibco), and serum-free B-27 Supplement (17504044; Gibco). 200 cells were seeded per well in 40 µl gastruloid N2B27 medium in Corning 96-Well Clear Ultra Low Attachment Microplates (7007; Corning) and incubated at 37°C, 5% $CO_2$. 48 h after seeding, 150 µl of gastruloid N2B27 medium with 3 µM GSK3β inhibitor CHIR99021 was added to each well. After 24 h, the Chiron-pulse was stopped by replacing 150 µl of Chiron-medium with 150 µl of basic gastruloid N2B27 medium. The medium was changed every day and images acquired using the ZOE Fluorescent Cell Imager (Bio-Rad).

## Gastruloid fixation and immunostaining

Gastruloids were prepared for imaging as previously described (Beccari et al, 2018). Before the staining procedure, plates were coated in gastruloid blocking buffer PBS-FT (PBS, 10% FBS, 0.2% Triton) to avoid attachment of gastruloids. All pipette tips were cut and pre-coated in PBS-FT solution to ensure safe gastruloid transfer without damage or loss. Gastruloids differentiated for 48, 72 and 96 h in Corning 96-Well Clear Ultra Low Attachment Microplates (7007; Corning) were then transferred to a six-well plate with 2 ml PBS. All replicate wells of the same condition were pooled in one well. To enable easier transfer, the plate was moved in a circular manner until all gastruloids accumulated in the center of each well. They were collected with a 1 ml pipette, kept in a vertical position to enable gastruloid sedimentation at the bottom of the tip. The tip was slightly pushed and the "hanging drop," containing mainly gastruloids with minimal carry-over, was transferred to a new well containing 2 ml of 4% PFA. Samples were incubated for 2 h at 4°C. Gastruloids were washed 3x with PBS-FT, each washing step containing a transfer step into a new well and remained for 10 min in the last well. Samples were either stored at 4°C until further use or the staining procedure was continued. They were transferred to a

12-well plate and blocked in 2 ml PBS-FT for 1 h on an orbital shaker. To reduce antibody volumes, gastruloids were placed in a 48-well plate containing 120 µl primary antibody solution and incubated for 24 h on an orbital shaker at 4°C. Samples were washed 3x with PBS-FT, including a 20 min incubation time of the last washing step and stained in 120 µl secondary antibody solution and DAPI staining (2.5 µg/ml) while shaking overnight at 4°C. After washing gastruloids 3x in PBS-FT, they were mounted. Cleaning of microscope slides with ethanol ensured removal of dust particles. A drop of 20 µl of VECTASHIELD Antifade Mounting Medium with DAPI (VEC-H-1200; Vector Laboratories) was placed to the center of a microscope slide and a small drop to the center of the cover slip to avoid air bubbles. Up to 10 gastruloids (depending on the time point and their size to avoid clustering) were placed within each drop on the microscope slide and carefully covered with a cover slip. After removal of excess liquid, the border of the cover glass was sealed with several layers of nail polish and all specimens stored at 4°C until imaging.

## Generation of KO cell lines

Cilium and centriole KO cell lines were generated with CRISPR/Cas9 (Cong et al, 2013). Forward and reverse DNA oligonucleotides were designed in Benchling, containing the gRNA-Sequence to the target gene as well as the overhangs required for cloning and were synthesized by Microsynth AG. Two guides targeting one exon and the following intron of the respective gene locus of interest were designed for each KO cell line. Forward and reverse DNA oligonucleotides were annealed and inserted into the vector plasmid pX330-U6-Chimeric_BB_CBh_hSpCas9 (42230; Addgene), using BbsI-HF (R3539L; NEB) directed cloning. The sequence integrity was confirmed by Sanger sequencing. One day before transfection, 100,000 R1 mESCs per well were seeded in 12-well plates and the medium of the cells was replaced on the next day 1 h before transfection. Plasmid combinations, containing 2 sgRNAs (700 ng each) targeting one exon and one intron of each gene, were co-transfected with the fluorescent marker plasmid dsRed (100 ng)—as a proxy for transfection efficiency—using Lipofectamine 2000 Transfection Reagent (11668019; Invitrogen). The medium of transfected cells was changed after 6–12 h. Two to three days after transfection, single dsRed+ cells were FACS-sorted (BD FACSMelody Cell Sorter) onto a fibronectin-coated (10 µg/ml) 96-well plate to enable the generation of single clone KO cell lines. Successful KO of respective genes was confirmed by genotyping PCRs, mapping primers outside the deleted region to obtain a smaller fragment and combining outside-inside primers, resulting in a PCR product in the WT (parental cell line) but not in the KO. Genotyping was performed first with direct lysis reagent DirectPCR Lysis (VIAG302-C; Viagen Biotech) according to the data sheet to select clones for expansion. After clone selection and expansion based on genotyping, genomic DNA was extracted with the Puregene Core Kit A (158043; QIAGEN) and genotyping PCRs were repeated and samples analyzed via Sanger sequencing (Figs S2 and S6). In addition, genome editing was

confirmed in immunofluorescence staining followed by imaging and Western blotting, if applicable.

## Generation of *Plk4* KO-*TP53* KO cell lines

*Plk4* KO cell lines were generated as described in the chapter "Generation of KO Cell Lines." Subsequently, p53 KO was introduced into these cell lines using the same CRISPR/Cas9 strategy. After clone selection and expansion based on genotyping, genomic DNA was extracted with the Puregene Core Kit A (158043; QIAGEN) and genotyping PCRs were repeated and samples analyzed via Sanger sequencing. In addition, genome editing was confirmed in immunofluorescence staining followed by imaging.

## Generation of *Cep83*/*Cep83Δexon4* overexpression cell lines

For the generation of *Cep83*/*Cep83Δexon4* overexpression rescue cell lines, we cloned PiggyBac (pB) expression plasmids, containing *Cep83* or *Cep83Δexon4* RNA, isolated from WT (parental cell line) or *Cep83Δexon4* cell lines, respectively, under an EF1α promoter. In addition, a N-terminal Flag-tag was added. Transfection of pB-Ef1α-Flag-Cep83-Ubi-Puro or pB-Ef1α-Flag-Cep83Δexon4-Ubi-Puro was used by Lipofectamine 2000 Transfection Reagent, using 500 ng of the construct and 1 μg PiggyBac transposase.

## RT-qPCR analysis

Cells were either starved according to the chapter "Cell starvation (2D)" or cultivated in 2iLIF mESC medium (KO cell lines, WT control) or under differentiation conditions (N2B27 without 2iLIF for 48 h) for the differentiation control. The RNA of the cells was purified using phenol–chloroform extraction, followed by isopropanol precipitation and a 75% ethanol wash, followed by DNAseI treatment according to the manufacturer's TriFast protocol (peqlab). Reverse transcription was performed with 1 μg of RNA using the SensiFAST cDNA Synthesis Kit (Meridian), following the manufacturer's protocol. RT-qPCR was conducted using the SYBR mastermix provided by the Vienna BioCenter facility. *Rpl13a* was used as a housekeeping gene. For analysis, ΔΔCt values were calculated and depicted as fold change relative to non-starved cells in a $\log_2$ scale. Statistics were calculated based on ΔCt values and tested for significance using an unpaired *t* test.

## Western blot analysis

For Western blotting, cells were detached with trypsin from six-well plates, the cell pellet washed with PBS and stored at –80°C after removal of the supernatant. Cells were lysed in 40 μl 1x RIPA buffer (20-188; Millipore) containing cOmplete Protease Inhibitor Cocktail (11836145001; Sigma-Aldrich) for 1 h on ice and vortexed every 10 min. Whole-cell extracts were collected by centrifugation (16,000*g*, 10 min, 4°C), cell debris removed and the protein concentration quantified using the Protein Assay Dye Kit (500-0006; Bio-Rad). 30 μg of protein per sample was resolved on a 10% SDS-polyacrylamide gel and transferred to a PVDF membrane (88518; Thermo Fisher Scientific), using the Wet/Tank Blotting System (Bio-Rad) at 400 mA for 1 h at 4°C. After Ponceau S staining of the

membranes, they were washed in TBS (1x Tris-buffered saline with 0.1% Tween 20) and blocked with 5% milk in TBST for 30 min. Primary antibody incubation was performed overnight at 4°C, the secondary HRP conjugated antibodies incubated for 1 h at RT, and the signal was detected with the Amersham ECL Select Western blotting Detection Reagent (RPN2235; Cytiva).

## Propidium Iodide (PI) staining and FACS analysis

For cell cycle analysis, 150,000 cells were seeded in six-well plates in 2iLIF in parallel to the mESC rosette cell assay. After 48 h, cells were detached with trypsin and counted. Cells were centrifuged for 5 min at 300*g*. Pellets with 500,000 cells per condition were first resuspended in 150 μl PBS to avoid clumping and 350 μl of 100% ice-cold ethanol was added dropwise while vortexing (final concentration 70% ethanol). Samples incubated for at least 30 min or overnight at 4°C and centrifuged for 5 min at 300*g*. For PI staining, pellets were resuspended in 300 μl PI-RNAse solution, containing 50 μg/ml PI (P4170; Sigma-Aldrich) and 100 μg/ml RNAse A (EN0531; Thermo Fisher Scientific), diluted in PBS and incubated for 20 min in the dark at RT. Single cell suspensions were obtained by transferring samples through a cell strainer cap (Corning) and measured with the LSRFortessa High-Parameter Flow Cytometer (BD Life Sciences—Biosciences). Data were analyzed using the FlowJoTM software (BD Life Sciences—Biosciences, version 10.5.3).

## Microscope specifications

Images were acquired with the following microscopes: Visitron Live Spinning Disk, Spinning disc units: Yokogawa CSU-X1-A1 spinning disk (50 μm pinholes, spacing 253 μm, 1,700*g*), Yokogawa CSU-W1-T2 spinning disk (50 μm pinholes, spacing 500 μm, 1,000*g*), Cameras: EM-CCD (back-illuminated Andor iXon Life 888, 1,024 × 1,024 pixel, 13 μm pixel size, 16 bit, 26 fps [full frame], QE > 95%) and sCMOS (back-illuminated Teledyne Prime BSI, 2,048 x 2,048 pixel, 6.5 μm pixel size, 16 bit, 43 fps [full frame], QE >95%), Objectives: CFI Plan Apo λ S 40xC/1.25 Sil, WD 0.30 mm (with coverslip thickness correction collar), CFI Plan Apo λ 60x/1.42 Oil, WD 0.15 mm, CFI Plan Apo λ 100x/1.45 Oil, WD 0.13 mm, Software VisiView 6.0, Visitron Spinning Disk, Spinning disk unit: Yokogawa CSU-X1 Nipkow spinning disk unit (50 μm pinholes, spacing 253, 1,700*g*), Camera: sCMOS (70% QE, 2,048 × 2,048 pixel, 6.45 pixel size, 16 bit, up to 100 fps), Objective: Plan-Apochromat 63x/1.4 Oil DIC, WD 0.19 mm, Software VisiView 6.0.

Zeiss LSM 980, Scanning mirrors with up to 8,192 × 8,192 pixels, Airyscan 2 compound detector consisting of 32 GaAsP detector units, Objective: Plan-Apochromat 63x/1.4 Oil DIC M27 (WD 0.19 mm), Software Zeiss ZEN 3.3.

## Image processing and data analysis

Acquired images were analyzed using Fiji (Schindelin et al, 2012). Gastruloid growth and elongation was quantified based on morphology of images acquired daily using the ZOE Fluorescent Cell Imager (Bio-Rad). Perimeter, roundness (roundness = 4pi [area/perimeter^2]), Feret's diameter, and length were quantified in gastruloids by taking manual measurements with the polygon and

segmented line tool (Fig S5C). Cilium length was determined by measuring the length of fluorescence for the ciliary marker ARL13B, ciliation rate (%) calculated based on counted cilia and DAPI staining to determine the cell number of the respective, quantified image. In gastruloids, the mean ratio of cilia to centrioles was determined based on immunofluorescence staining of gastruloids labeled with antibodies against ARL13B and *γ*-tubulin. This quantification method is not a direct readout of the ciliation rate per cell, due to difficulties in attributing centriole signal to individual cells in 3D samples, and was therefore only used to compare ciliogenesis in those samples. In 3D in vitro rosettes, we quantified lumen and rosette size based on masking the central z-plane (in-house developed Python code), using PODXL as a lumen marker and DAPI to assess the rosette size. Imaging conditions and subsequent post-acquisition processing was always constant within each experiment for image analysis. For representative visual depiction in Figures, images were cropped, brightness, and contrast optimized using Fiji.

### Graphing and statistics

All graphs were created using GraphPad Prism 10 and statistically analyzed using a two-tailed unpaired *t* test (normal distributed data) for independent data sets including at least three experimental replicates. *P* values: ns > 0.05, *$P \le 0.05$, **$P \le 0.01$, ***$P \le 0.001$, ****$P \le 0.0001$.

# Data Availability

All data, cell lines and reagents generated and described in the manuscript will be shared upon request.

# Supplementary Information

# Acknowledgements

We would like to thank all members of the Buecker lab for discussions and feedback through the project, including the summer student Laura Rüland for her help, the Dammermann and Leeb labs for shared lab meetings/ journal clubs, critical input and valuable support, the BioOptics-FACS and BioOptics-Light Microscopy facility at Max Perutz Labs, Nicholas Wedige and Lorenz Perschy for help with the image analysis pipeline of 3D in vitro rosettes. This work was supported by the Austrian Science Fund FWF (PAT9017923, P34123, W1261 and DOC72 to C Buecker and F8803-B to A Dammermann).

### Author Contributions

I Voelkl: conceptualization, data curation, investigation, and writing—original draft.
T Civetta: data curation, formal analysis, and writing—review and editing.
M Egg: data curation.
M Huber: data curation and formal analysis.
S Feng: data curation and formal analysis.
A Dammermann: conceptualization and writing—review and editing.
C Buecker: conceptualization, funding acquisition, investigation, project administration, and writing—review and editing.

### Conflict of Interest Statement

The authors declare that they have no conflict of interest.

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
