## [Reviewer comments · Life Science Alliance]

In vitro approaches to study centriole and cilium function in early mouse embryogenesis

Isabella Voelkl, Tamara Civetta, Mirijam Egg, Marie Huber, Songjie Feng, Alexander Dammermann, and Christa Buecker
DOI: <https://doi.org/10.26508/lsa.202503358>

Corresponding author(s): Christa Buecker, Max Perutz Labs

Review Timeline:

Submission Date:	2025-04-11
Editorial Decision:	2025-05-22
Revision Received:	2025-09-02
Editorial Decision:	2025-09-30
Revision Received:	2025-10-09
Accepted:	2025-10-13

Scientific Editor: Sarita Hebbar

Transaction Report:

May 23, 2025

Re: Life Science Alliance manuscript #LSA-2025-03358-T

Dr. Christa Buecker
University of Vienna
Max Perutz Labs
Dr Bohrgasse 9
Vienna 1030
Austria

Dear Dr. Buecker,

Thank you for submitting your manuscript entitled "In vitro approaches to study centriole and cilium function in early mouse embryogenesis" to Life Science Alliance. The manuscript was assessed by three expert reviewers, whose comments are appended to this letter.

All three reviewers commented that this work will be of potential use to the community. However we agree with the reviewers that the manuscript needs additional data and discussion points for publication at LSA.

A revised manuscript must include the following:

1. Quantitative data to demonstrate the robustness of the observed phenotypes, and clear indication of sample size, number of independent experiments analysed, and statistical significance as suggested by all the reviewers (Reviewer 1, various points-- under Figure 2, 4, 5,7,8 and Reviewer 2, major point 2, and Reviewer 3, major point 2). Reviewer 2 also specified that an accurate description of the cilia phenotype of Cep83 exon 4 deletion mutant in 2D and 3D models needs to be clearly stated. Here, we do not expect further experiments on subcellular localization of the mutant CEP83 protein and basal body maturation demonstrated by recruitment of CEP164 and removal of CCP110
2. A more detailed characterisation of the fate of the starved cells. We leave it to the authors' choice to follow any of the suggestions provided by Reviewer 1, point on figure 1.
3. Expression of pluripotency/differentiation and cell cycle/cell death markers required for WT vs KO (Ift88/Plk4) (Reviewer 1, point on Figure 3 and Reviewer 2, major point 1).
4. Validation by western blotting to demonstrate absence of protein or presence of any truncated protein (Ift88/Plk4,Cep83) in the derived KO clones (Reviewer 1, points under figure 3 and 7)
5. Discussion on potential factors, other than polarisation, contributing to differences between 2D and 3D cultures (Reviewer 3, point 1).
6. Discussion on the in-vivo equivalence of the lumen of gastruloids, and on the identity of cells surrounding the lumen (Reviewer 3, point 2).

In line with the overall recommendations, we invite you to submit a revised manuscript addressing the reviewers' comments. While a rebuttal must respond to all points in some form, additional data to resolve these points (other than ones indicated above) is not required.

Thank you for this interesting contribution to Life Science Alliance. We are looking forward to receiving your revised manuscript.

Sincerely,

Sarita Hebbar, PhD
Scientific Editor
Life Science Alliance

B. MANUSCRIPT ORGANIZATION AND FORMATTING:

Reviewer #1 (Comments to the Authors (Required)):

1. Summary of the findings of the manuscript and their importance

The manuscript by Isabella Voelkl and colleagues examines the role of centrioles and cilia regulators in early embryonic events using mouse embryonic stem cell cultures, rosettes, and gastruloids, in combination with gene editing. They report that both centrioles (Plk4 KO) and cilia (Ift88 and Cep83 KO) regulators are dispensable for mESCs and rosettes derived from those. Furthermore, they show that ablation of Plk4 prevents the growth of gastruloids and leads to their disintegration. Finally, the report that deletion of exon 4 in Cep83 is dispensable for ciliogenesis in 3D cultures, in contrast to 2D models. "

The manuscript offers some insight into the role of cilia and centriole regulators in early mouse embryogenesis, and the use of 3D culture models has not been reported in that perspective before. However, the manuscript brings only incremental advancement in the understanding of the role of cilia/centrioles in early embryonic development over the current literature, and is very descriptive. Nevertheless, it could still serve as a useful resource for readers who want to get an initial insight into the utilization of 3D culture models for studying centrioles and cilia. This, however, requires a more thorough characterization of the derived models and described phenotypes, as outlined in my comments below.

2. Key findings of the manuscript

Figure 1

The observations are intriguing, but preliminary. The authors should provide data that demonstrates the efficiency of their protocol for the transition of mESCs into EpiLC cell fate. Moreover, to justify the conclusion that starvation, but not differentiation, drives ciliogenesis, it is essential to demonstrate that starvation is not causing mESCs to differentiate. In fact, the strong upregulation of protein levels of CDK inhibitor p27 suggests a diversion from the undifferentiated state, in line with reports on the tight connection of cell cycle regulation and undifferentiated state/pluripotency regulation (reviewed in PMID: 26410405) or the reported role of p27 in repressing a core pluripotency gene Sox2 (PMID: 23217425). Besides examining a set of pluripotency/differentiation makers to assess the fate of the starved cells (which should be done in any case), a washout experiment could be considered to demonstrate eventual reversibility of the phenotype (to test whether ciliogenesis falls back following 2iLIF re-addition).

Figure2

The included data provide insight into the establishment of in vitro rosette differentiation, which is subsequently used later. The presented data on cilia formation are limited, showing representative images, no quantification of the incidence of ciliogenesis is

provided (while the incidence of cilia formation is measured in gastruloids in Fig5). This should be done, either as part of this Figure, or to support conclusions of Figure 4 (see later). This drawback (lack of quantification/information regarding sample size) also echoes in my next comments.

Figure 3

Similarly to Figure 1, the reported findings are intriguing, but the conclusions are preliminary. Thorough characterization of the derived KO clones is needed (e.g., WB to demonstrate absence of Ift88/Plk4 protein or any truncated variant, expression of pluripotency/differentiation markers in WT vs KO).

Figure 4

As already mentioned, the conclusion regarding the absence of cilia and/or centrioles or induction of PODXL expression is now entirely based on representative images of one 3D rosette per each condition. This needs to be supported by quantifications to demonstrate the robustness and reproducibility of the phenotype (or the lack of phenotype in the case of PODXL expression).

Figures 5

I appreciate the provided assessment of cilia formation incidence, which supports the conclusion of this figure. Please provide information on the sample size and number of independent experiments analysed. This is pertinent to all quantifications included in the manuscript, as information on how many cells/rosettes/gastruloids were examined (and in how many experiments) is lacking.

Figure 6

The conclusion regarding the role of centrioles in gastruloid growth would require additional means to ablate centriole formation, independent of Plk4 (as the authors did for cilia, by targeting IFT88 or CEP83). As this would require substantial additional work, I suggest toning down the conclusions - the phenotypes can be convincingly linked to Plk4 ablation; however, whether they represent a direct consequence of loss of the kinase per se or loss of centrioles cannot be unambiguously deciphered from the provided data.

Figure 7

Analogous to my previous comments, provide means of demonstrating the absence of Cep83 protein (and presence of truncated version lacking the exon 4) in derived KO clones, and quantification of the cilia defect phenotype.

Figure 8

The conclusion regarding dispensability of exon4 of CEP83 for ciliogenesis in 3D culture is well demonstrated in the included data (provided the information regarding sample size is added and the expression of truncated CEP83 protein is confirmed).

3. Additional recommendations

Previous work has examined the impact of centriole or cilia ablation in mouse or human embryonic stem cell cultures. If possible, the work of colleagues should be acknowledged (cited), and the manuscript findings discussed in light of these reports.

PMID: 30197118 - depletion of centrioles by PLK4 inhibition in hPSCs

PMID: 33410253- depletion of centrioles in mESCs following Sas4 ablation

PMID: 35478224 - depletion of cilia in hPSCs following Kif3a/b ablation

PMID: 37558899- depletion of cilia in hPSCs by TTBK2 ablation.

The numbering of supplementary figures should ideally reflect the order in which they are mentioned in the text (right now, Suppl Figure 4 is mentioned first in the manuscript, which is confusing - why not make it Suppl. Figure 1?).

Reviewer #2 (Comments to the Authors (Required)):

Voelkl et al. employed two 3D in vitro systems derived from mouse embryonic stem cells (mESCs) to investigate the role of centrioles and cilia during early development. These 3D models closely mimic the peri-implantation and early gastrulation stages of embryogenesis, up to embryonic day 8.5 (E8.5). The authors demonstrate that while cilia are dispensable for early embryonic patterning, defects in centriole assembly can lead to early arrest in gastruloid development-independent of cilia formation. This work nicely aligns with previous in vivo studies (Bazzi & Anderson, 2014; Bang et al., 2015).

In my view, the most intriguing finding is that cilia formation in the 3D model appears capable of partially compensating for defects in CEP83 function, particularly in the Cep83 Δ exon4 mutant.

Before recommending publication, I suggest the authors address the following concerns:

Major concerns:

1. Centriole Loss and Cell Death in Plk4 Mutants:

The study shows that Plk4 mutant mESCs display no apparent cell cycle defects in 2D but fail to form gastruloids in 3D. Given that centriole loss can trigger p53-dependent cell death (as shown in Sas6 mutants), and considering that PLK4 and SAS6

function in parallel pathways, the authors should examine: Cell cycle progression and cell death markers and p53 expression levels

These analyses should be performed either at the 48-hour aggregation stage or within the 3D rosette structures. While Fig. 3D indicates normal proliferation in monolayer culture, the critical context here is the 3D environment, where cellular reorganization is actively occurring.

2. Characterization of Cep83 Δ exon4 Function:

The observation that Cep83 Δ exon4 mutants can form cilia at later stages is intriguing but lacks mechanistic clarity. Specific concerns include: The subcellular localization of the mutant CEP83 protein; Whether basal body maturation occurs normally (e.g., recruitment of CEP164 and removal of CCGP110, etc.); It should be revised to accurately state that cilia do form in the Cep83 Δ exon4 background, as shown in Fig. S5 the mutant cells in 2D differentiation do form cilia; Throughout the manuscript, cilia phenotypes must be described with more precision-including frequency and length.

This applies to figures such as Fig. 2D, Fig. 3C, Fig. 4A, Fig. 7C-D, and Fig. 8A-B, where the use of "+" or "-" is insufficient. Quantitative data should be included either in the figure or text, ideally supported by a dot or box plot.

Minor concerns:

In figures showing gastruloid roundness over time (e.g., Fig. 6C, Fig. 7E-F), please clarify how many gastruloids were quantified across the three biological replicates. Also, the use of outlier dots without scatter can make interpretation difficult.

Statistical significance is not indicated in several key figures (e.g., Fig. 5B, 6C, 7E-F, 8C, S2B). This should be added for clarity and scientific rigor.

Avoid using green/red combinations due to accessibility issues for colorblind readers. In addition, remove DAPI or add arrows when labeling cilia or centrioles to improve figure clarity (e.g., Fig. 3C, Fig. 7C-D).

Please specify the genotype of the wild-type (WT) control. Is it a parental mESC line or one with scramble sgRNA? Also, some plots appear redundant-for example, showing both box plots and bar plots of the same data (e.g., Fig. S2B, S3B). Consider consolidating them for clarity.

Reviewer #3 (Comments to the Authors (Required)):

In this manuscript, Voelkl et al investigate the role of centrioles and cilia during early post-implantation development using different in vitro stem cell models. Their results highlight the importance of a 3D environment for cilium formation. Moreover, they show that while centrioles are essential for gastruloid formation, the lack of cilia only causes minor defects. This is a well-conducted study, very clearly presented, and with well-supported conclusions. I have the following suggestions to improve the quality of the manuscript:

1. Difference between 2D and 3D: In the discussion, the authors mention that the key difference between 2D and 3D is polarisation, but this is not necessarily the case. First, polarisation could be induced in 2D by coating the dish with Matrigel or laminin instead of fibronectin. Would this lead to cilia formation in EpiLCs? If not, other variables such as the stiffness of the substrate or membrane curvature may be responsible for the 2D vs 3D differences.

2. Gastruloid experiments:

- The authors show that cilia appear in areas of the gastruloids that have lumens. What is the in vivo equivalence of these lumens? What is the identity of the cells that surround these lumens? As far as I know, gastruloids do not have an equivalent of the pro-amniotic cavity.

- What is the % of cells with cilia at each step of the gastruloid protocol? From the data, it appears that only a small % of cells have cilia after the CHIR pulse. If this is the case, then the initial pool of pluripotent cells before the pulse (equivalent of E5.5 epiblast) does not have cilia, which would not mimic what happens in the embryo. Could it be that the lack of effects observed in terms of differentiation and lineage specification are a consequence of the lack of cilia in the primed pluripotent cells in this model? If this is the case, then alternative in vitro models could be considered. Rosettes can be used as a starting point to study early primitive streak differentiation (Harrison et al, Science, 2017, and Satoh et al, Developmental Cell, 2024). This should be discussed.

Minor comments:

- Figure 1A: The drawing of the blastocyst is not correct; there are too many epiblast cells.

RESPONSE TO REVIEWERS

We would like to thank all three reviewers for their time and their constructive feedback on our manuscript. In preparing this revision, we have sought to incorporate their comments as far as possible and included additional data to support our claims. We believe our manuscript to be significantly strengthened by these additions and hope the reviewers will find our work now suitable for publication in *Life Science Alliance*.

Below we first outline the major changes that have been made to the manuscript and then address the individual points raised by each reviewer.

Summary of major changes

1) *Additional quantitative data to demonstrate the robustness of the observed phenotypes and indications of sample size, number of independent experiments analyzed and statistical significance.*

We have now incorporated this additional information into each figure legend and into the text where applicable. In general, we always analyzed at least three independent experiments and generated at least two independent clonal cell lines for each experiment described in this manuscript.

2) *Further characterization of the fate of starved cells.*

We now include additional experiments in Supplemental Figure S1 to compare the differentiation status of cells starved for either 24 h or 48 h after nutrient withdrawal using qPCR with primers against core pluripotency markers (*Oct4* and *Sox2*), naive pluripotency markers (*Tbx3*, *Klf4* and *Esrrb*) and formative pluripotency markers (*Fgf5*, *Oct6* and *Otx2*). Neither at 24 h nor at 48 h of starvation was a change in the expression of these markers observed. We can therefore conclude that while ciliation increases during starvation, this is not due to a change in differentiation status.

3. *Examining the effect of KOs on pluripotency status/differentiation and cell cycle progression/cell death*

We now include qPCRs in Supplemental Figure S3 to demonstrate that none of our KOs induce premature differentiation in the ESC state and all are fully naive pluripotent. In the original manuscript, we had speculated that the strong phenotype of *Plk4* KO in gastruloids might be due to activation of the mitotic surveillance pathway. If so, a double knockout of *Plk4* and p53 should rescue this phenotype. We have now included additional experiments in our revised manuscript (see Figure 6 A and S5 A, B) to address this question. Indeed, co-deletion of p53 in *Plk4* KO cells resulted in a rescue of the severe phenotype in gastruloids. However, a more comprehensive analysis of the phenotype of these double mutants is beyond the scope for this manuscript.

4. *Validation of loss of protein in KO clones.*

We recognize this is an important point and we would have liked to include validation by immunoblots for all three proteins, PLK4, CEP83 and IFT88.

We now include blots for IFT88, demonstrating that the protein is indeed lost from KO cells (see Supplemental Figure S2 C).

We were unable to show comparable blots for PLK4. This is not entirely unexpected, as PLK4 is typically expressed at very low levels. Consequently, most studies on PLK4, including those examining its autoregulation (e.g., Holland et al., J Cell Biol 2010), have relied on exogenous overexpression of epitope-tagged transgenes (eg Holland et al., J Cell Biol 2010). In our hands, commercial but also home-made antibodies against PLK4 failed to show convincing signal by immunoblotting of wild type mESCs. However, the total loss of centrioles in *Plk4* KO cells along with the results of our genotyping PCRs gives us confidence that PLK4 function is perturbed.

In the case of CEP83, we faced similar issues of detection. Of the three commercial antibodies tested (Sigma HPA038161, Invitrogen PA576401, Proteintech 26013-1-AP), only the third detected a specific band in lysates of WT cells and even then weakly. That band was seemingly reduced in both full KO and CEP83exon4 deletion, although background bands complicate interpretation (see Rebuttal Figure R1, below). We further conducted qPCRs with probes to exons 2 and 3, which should remain in any truncated mRNAs in the full KO/exon 4 deletion. In the full KO, signal is reduced 5-10-fold, consistent with nonsense-mediated decay of the frameshifted transcript in that mutant (Figure R1). In the Δ Exon4 cell lines, transcript levels were only slightly reduced, as expected for an in-frame deletion. While these results are consistent with the observed differences in phenotypic severity between the two mutants, we prefer not to include such ambiguous data in our manuscript.

5. Additional discussion

Beyond the inclusion of the above experimental data and documentation, we have revised the text to include a discussion of potential factors besides polarization that could account for the observed differences between 2D and 3D cultures, as well as the *in vivo* equivalent of the lumen of gastruloids, and the identity of cells surrounding the lumen.

Reviewer #1

1. Summary of the findings of the manuscript and their importance

The manuscript by Isabella Voelkl and colleagues examines the role of centrioles and cilia regulators in early embryonic events using mouse embryonic stem cell cultures, rosettes,

and gastruloids, in combination with gene editing. They report that both centrioles (Plk4 KO) and cilia (Ift88 and Cep83 KOs) regulators are dispensable for mESCs and rosettes derived from those. Furthermore, they show that ablation of Plk4 prevents the growth of gastruloids and leads to their disintegration. Finally, the report that deletion of exon 4 in Cep83 is dispensable for ciliogenesis in 3D cultures, in contrast to 2D models. "

The manuscript offers some insight into the role of cilia and centriole regulators in early mouse embryogenesis, and the use of 3D culture models has not been reported in that perspective before. However, the manuscript brings only incremental advancement in the understanding of the role of cilia/centrioles in early embryonic development over the current literature, and is very descriptive. Nevertheless, it could still serve as a useful resource for readers who want to get an initial insight into the utilization of 3D culture models for studying centrioles and cilia. This, however, requires a more thorough characterization of the derived models and described phenotypes, as outlined in my comments below.

We thank the reviewer for their time and expertise in evaluating our manuscript. We agree that the work included in our manuscript is mostly descriptive, but we believe that the murine *in vitro* models we describe will be of great interest to the community as a starting point to further study the role of centrioles and cilia in mammalian early development.

2. Key findings of the manuscript - Figure 1

The observations are intriguing, but preliminary. The authors should provide data that demonstrates the efficiency of their protocol for the transition of mESCs into EpiLC cell fate.

The transition from ESC to EpiLCs is a highly efficient cell fate transition that we and others have used to study the regulation of cell fate transitions and enhancer mechanisms. The protocol we use here has been extensively characterized in previous studies from the lab (eg Buecker et al., Cell Stem Cell 2014; Thomas et al., Mol Cell 2021; Romeike et al., EMBO Rep 2022). Other labs have adopted similar differentiation strategies (eg Chen et al., Cell Stem Cell 2018; Boileau et al., Genome Biol 2023; Huth et al., Genes Dev 2022). Therefore, we have not included additional demonstrations of the efficiency of this protocol but instead refer the reader to those studies for further detail.

Moreover, to justify the conclusion that starvation, but not differentiation, drives ciliogenesis, it is essential to demonstrate that starvation is not causing mESCs to differentiate.

We thank the reviewer for this comment. We now include data in Supplemental Figure S1 to demonstrate that starvation in mouse embryonic stem cells does not induce differentiation. We compared the differentiation status of cells starved for either 24 h or 48 h after nutrient withdrawal using qPCR with primers against core pluripotency markers (*Oct4* and *Sox2*), naive pluripotency markers (*Tbx3*, *Klf4*, and *Esrrb*) and formative pluripotency markers (*Fgf5*, *Oct6*, and *Otx2*). Neither at 24 h nor at 48 h of starvation was a change in expression of these markers observed. Therefore, we conclude that while ciliation increases during starvation, this is not due to a change in differentiation status.

In fact, the strong upregulation of protein levels of CDK inhibitor p27 suggests a diversion from the undifferentiated state, in line with reports on the tight connection of cell cycle regulation and undifferentiated state/pluripotency regulation (reviewed in PMID: 26410405) or the reported role of p27 in repressing a core pluripotency gene Sox2 (PMID: 23217425). Besides examining a set of pluripotency/differentiation makers to assess the fate of the starved cells (which should be done in any case), a washout experiment could be considered to demonstrate eventual reversibility of the phenotype (to test whether ciliogenesis falls back following 2iLIF re-addition).

We thank the reviewer for raising this important point. As it happens, my lab recently released a preprint that examined the question of how heterogeneity is introduced during differentiation, and we focused on the cell cycle as one potential underlying mechanism that might allow some cells to differentiate earlier than others. However, we were unable to find such a strict relationship, and in our hands, a relationship between cell cycle and differentiation is not observed during the exit from naive pluripotency (see <https://doi.org/10.1101/2023.09.15.557731>). Another recent study suggests that metachronous exit from naive pluripotency might be caused by ERK signalling (Mulas et al., Development 2024). Finally, Allon Klein's lab demonstrated that cell state transitions in the zebrafish embryo are decoupled from cell divisions (Kukreja et al., Nat Cell Biol 2024). When the group abrogated cell division altogether, the embryo still developed and underwent cell state transitions despite the lack of different cell cycle phases. While the cell cycle can certainly play a role in differentiation decisions, these connections may be less critical than previously stated, something that should be revisited in future studies.

- Figure 2

The included data provide insight into the establishment of in vitro rosette differentiation, which is subsequently used later. The presented data on cilia formation are limited, showing representative images, no quantification of the incidence of ciliogenesis is provided (while the incidence of cilia formation is measured in gastruloids in Fig5). This should be done, either as part of this Figure, or to support conclusions of Figure 4 (see later). This drawback (lack of quantification/information regarding sample size) also echoes in my next comments.

We have now included quantifications of the incidence of cilia in the rosette assays in Figure 2D. We were not able to measure ciliary length to our standard in rosettes due to the small lumen, the extrusion of the cilia in different directions and the lower numbers of ciliated cells.

- Figure 3

Similarly to Figure 1, the reported findings are intriguing, but the conclusions are preliminary. Thorough characterization of the derived KO clones is needed (e.g., WB to demonstrate absence of Ift88/Plk4 protein or any truncated variant, expression of pluripotency/differentiation markers in WT vs KO).

As described above, we would have liked to validate all our KOs using immunoblots. However, with the exception of IFT88, this was not possible in mouse ES cells due to the low level of expression of those proteins and the lack of reagents capable of detecting the mouse protein at the required level of sensitivity. We hope that the analysis of different

independently derived clones verified by Sanger Sequencing and the phenotypic analysis are sufficient for the validation of these clones. Unfortunately, a Mass Spec analysis is beyond the scope for this revision.

As for the expression of pluripotency/differentiation markers, we have now included qPCRs demonstrating that the loss of *Plk4*, *Ift88* or *Cep83* does not affect naive pluripotency. We compared the differentiation status of different KOs to either WT cells in ESC condition or cells differentiated for 48 h into EpiLCs using qPCRs with primers against core pluripotency markers (*Oct4* and *Sox2*), naive pluripotency markers (*Tbx3*, *Klf4*, and *Esrrb*) and formative pluripotency markers (*Fgf5*, *Oct6*, and *Otx2*). None of the KOs show changes in expression of the different markers. Therefore, we conclude that KO of the selected genes does not affect the pluripotency state of the mESCs.

- *Figure 4*

As already mentioned, the conclusion regarding the absence of cilia and or/centrioles or induction of PODXL expression is now entirely based on representative images of one 3D rosette per each condition. This needs to be supported by quantifications to demonstrate the robustness and reproducibility of the phenotype (or the lack of phenotype in the case of PODXL expression).

The loss of centrioles and cilia in *Plk4/Ift88* mutants is total. Consistent with the critical role of those proteins in centriole assembly/ciliogenesis, centrioles (and cilia) are entirely absent in *Plk4* mutants, as are cilia (but not centrioles) in *Ift88* mutants. We have naturally performed all *in vitro* differentiation experiments multiple times, examining many fields of view for each condition. We now include additional information on our quantifications in the text and figure legends.

- *Figures 5*

I appreciate the provided assessment of cilia formation incidence, which supports the conclusion of this figure. Please provide information on the sample size and number of independent experiments analysed. This is pertinent to all quantifications included in the manuscript, as information on how many cells/rosettes/gastruloids were examined (and in how many experiments) is lacking.

We have added information on sample sizes, the number of analyzed independent clones, and the number of independent experiments throughout the manuscript, both in the figure legends and in the text.

- *Figure 6*

*The conclusion regarding the role of centrioles in gastruloid growth would require additional means to ablate centriole formation, independent of *Plk4* (as the authors did for cilia, by targeting *IFT88* or *CEP83*). As this would require substantial additional work, I suggest toning down the conclusions - the phenotypes can be convincingly linked to *Plk4* ablation; however, whether they represent a direct consequence of loss of the kinase per se or loss of centrioles cannot be unambiguously deciphered from the provided data.*

We chose *Plk4* as a means to test the differential requirement for centrioles vs cilia in the developmental processes modeled in our *in vitro* assays because it is an extremely well

characterized master regulator of centrosome duplication. Unlike other polo-like kinases of which it is a highly divergent member, *Plk4* has no known targets beyond centrioles and associated structures such as centriolar satellites, and all phenotypes so far ascribed to *Plk4* depletion or inhibition such as cytokinesis failure (Hudson et al., Curr Biol 2001) of failure to progress in the cell cycle (Wong et al., Science 2015) can be traced to loss of centrosomes and/or cilia. Previous work had shown that a lack of centrioles leads to the activation of the mitotic surveillance pathway, resulting in cell cycle arrest or apoptosis through the activation of p53. In the revised manuscript, we demonstrate that the phenotypes observed in our *Plk4* KOs can likewise be rescued through co-deletion of p53 (See Figure 6 A and Supplemental Figure S5 B). We therefore conclude that the phenotype observed in our *Plk4* KOs is indeed due to lack of centrioles.

- Figure 7

Analogous to my previous comments, provide means of demonstrating the absence of Cep83 protein (and presence of truncated version lacking the exon 4) in derived KO clones, and quantification of the cilia defect phenotype.

We now include further quantification of the CEP83 ciliary phenotype. As for confirming the absence of CEP83 protein and potential expression of a truncated version in the exon 4 mutant, as discussed above we were unable to reliably detect CEP83 in immunoblots. *Cep83* mRNA levels do appear to be strongly reduced in the null mutant based on RT-qPCR, presumably due to nonsense-mediated decay. However, as expected for an in-frame deletion, the exon 4 mutant shows little change.

- Figure 8

The conclusion regarding dispensability of exon4 of CEP83 for ciliogenesis in 3D culture is well demonstrated in the included data (provided the information regarding sample size is added and the expression of truncated CEP83 protein is confirmed).

We have added information on sample sizes in the revised manuscript.

3. Additional recommendations

Previous work has examined the impact of centriole or cilia ablation in mouse or human embryonic stem cell cultures. If possible, the work of colleagues should be acknowledged (cited), and the manuscript findings discussed in light of these reports.

PMID: 30197118 - depletion of centrioles by PLK4 inhibition in hPSCs

PMID: 33410253- depletion of centrioles in mESCs following Sas4 ablation

PMID: 35478224 - depletion of cilia in hPSCs following Kif3a/b ablation

PMID: 37558899- depletion of cilia in hPSCs by TTBK2 ablation.

We have added the citations in the appropriate place in the text. We would like to note that while these studies are certainly highly relevant, none of them utilized *in vitro* models designed to recapitulate early stages of mammalian development as was done here.

The numbering of supplementary figures should ideally reflect the order in which they are mentioned in the text (right now, Suppl Figure 4 is mentioned first in the manuscript, which is confusing - why not make it Suppl. Figure 1?).

We have revised the order of figures to ensure it matches their first appearance in the text.

Reviewer #2

Voelkl et al. employed two 3D in vitro systems derived from mouse embryonic stem cells (mESCs) to investigate the role of centrioles and cilia during early development. These 3D models closely mimic the peri-implantation and early gastrulation stages of embryogenesis, up to embryonic day 8.5 (E8.5). The authors demonstrate that while cilia are dispensable for early embryonic patterning, defects in centriole assembly can lead to early arrest in gastruloid development-independent of cilia formation. This work nicely aligns with previous in vivo studies (Bazzi & Anderson, 2014; Bang et al., 2015).

In my view, the most intriguing finding is that cilia formation in the 3D model appears capable of partially compensating for defects in CEP83 function, particularly in the Cep83 Δ exon4 mutant.

Before recommending publication, I suggest the authors address the following concerns:

We thank the reviewer for the overall positive assessment of our manuscript. We have incorporated many of the suggested changes and hope to have addressed the concerns raised by the reviewer.

Major concerns:

1. Centriole Loss and Cell Death in Plk4 Mutants:

The study shows that Plk4 mutant mESCs display no apparent cell cycle defects in 2D but fail to form gastruloids in 3D. Given that centriole loss can trigger p53-dependent cell death (as shown in Sas6 mutants), and considering that PLK4 and SAS6 function in parallel pathways, the authors should examine: Cell cycle progression and cell death markers and p53 expression levels

These analyses should be performed either at the 48-hour aggregation stage or within the 3D rosette structures. While Fig. 3D indicates normal proliferation in monolayer culture, the critical context here is the 3D environment, where cellular reorganization is actively occurring.

The reviewer has raised an important point and we have decided to address this point in depth by generating a dual knockout of *Plk4* and p53. In the new data, now included in Figure 6 A and Supplemental Figure S5 A, C, we show that co-deletion of p53 in *Plk4* KOs can rescue the observed phenotype, and gastruloids develop similarly to WT. These data demonstrate that, as previously observed for *Sas6* in mESCs (Grzonka and Bazzi, eLife 2024) and *Sas4* in mice (Bazzi and Anderson, PNAS 2014), silencing of the mitotic surveillance pathway can restore development of *Plk4* KO cells. Importantly, unlike for *Sas6*, centrioles do not form in *Plk4* mutants under any of the tested conditions, allowing us to assess the consequences of full centriole loss.

2. Characterization of Cep83 Δ exon4 Function:

The observation that Cep83 Δexon4 mutants can form cilia at later stages is intriguing but lacks mechanistic clarity. Specific concerns include: The subcellular localization of the mutant CEP83 protein; Whether basal body maturation occurs normally (e.g., recruitment of CEP164 and removal of CCP110, etc.); It should be revised to accurately state that cilia do form in the Cep83 Δexon4 background, as shown in Fig. S5 the mutant cells in 2D differentiation do form cilia; Throughout the manuscript, cilia phenotypes must be described with more precision-including frequency and length.

This applies to figures such as Fig. 2D, Fig. 3C, Fig. 4A, Fig. 7C-D, and Fig. 8A-B, where the use of "+" or "-" is insufficient. Quantitative data should be included either in the figure or text, ideally supported by a dot or box plot.

We agree with the reviewer that the phenotype of Cep83 Δexon4 mutants is highly intriguing. However, following the editor's advice we have not pursued a further mechanistic dissection of the mutant phenotype for the current manuscript. We do, however, clarify the residual ciliation in the exon 4 mutant under 2D conditions in the appropriate place in the text.

We have further added quantitative data in the text and figures as requested by the reviewer. It should be noted that with the exception of the Cep83 mutants, loss of centrioles and cilia in Plk4 mutants and loss of cilia in Ift88 mutants is total. Consistent with their known central role in centriole assembly/ciliogenesis, centrioles and/or cilia were never observed in those conditions. "-" serves to indicate that. This is now clarified in the text.

Minor concerns:

In figures showing gastruloid roundness over time (e.g., Fig. 6C, Fig. 7E-F), please clarify how many gastruloids were quantified across the three biological replicates. Also, the use of outlier dots without scatter can make interpretation difficult.

We have included this information in the figure legends.

Statistical significance is not indicated in several key figures (e.g., Fig. 5B, 6C, 7E-F, 8C, S2B). This should be added for clarity and scientific rigor.

We have added statistical tests wherever we make claims that require support by statistical analysis. The reviewer is referring to the gastruloids experiments here. Gastruloid differentiation tends to be highly variable and absolute numbers can vary drastically from experiment to experiment. Comparisons can therefore be performed only within a given experiment.

Avoid using green/red combinations due to accessibility issues for colorblind readers. In addition, remove DAPI or add arrows when labeling cilia or centrioles to improve figure clarity (e.g., Fig. 3C, Fig. 7C-D).

We thank the reviewer for this comment and will consider it for future manuscripts. We have included individual channels to make the figures accessible to colour blind readers. As suggested by the Reviewer, we have now added arrows to label cilia and centrioles.

Please specify the genotype of the wild-type (WT) control. Is it a parental mESC line or one with scramble sgRNA?

We now indicate this in each figure legend. In general, we always compare KOs to their parental mESCs.

Also, some plots appear redundant-for example, showing both box plots and bar plots of the same data (e.g., Fig. S2B, S3B). Consider consolidating them for clarity.

The reviewer is correct - some of the plots appear partly redundant. As noted above, gastruloids exhibit substantial variability. We were therefore very careful in comparing only experimental conditions that were directly comparable. This meant comparing each clone to a WT cell line grown in parallel on the same 96-well plate and prepared at the same time. This unfortunately also limits our ability to consolidate data as suggested by the reviewer.

Reviewer #3

In this manuscript, Voelkl et al investigate the role of centrioles and cilia during early post-implantation development using different in vitro stem cell models. Their results highlight the importance of a 3D environment for cilium formation. Moreover, they show that while centrioles are essential for gastruloid formation, the lack of cilia only causes minor defects. This is a well-conducted study, very clearly presented, and with well-supported conclusions. I have the following suggestions to improve the quality of the manuscript:

We would like to thank the reviewer for their overall positive assessment of our manuscript. We have incorporated the suggested changes where possible into our revised manuscript and hope that they address the concerns raised by the reviewer.

1. Difference between 2D and 3D: In the discussion, the authors mention that the key difference between 2D and 3D is polarisation, but this is not necessarily the case. First, polarisation could be induced in 2D by coating the dish with Matrigel or laminin instead of fibronectin. Would this lead to cilia formation in EpiLCs? If not, other variables such as the stiffness of the substrate or membrane curvature may be responsible for the 2D vs 3D differences.

We have now included additional discussion on the difference between 2D and 3D cultures; however, we have not included additional experiments to change the substrate and the stiffness of the substrate in our differentiation experiments. This would have required extensive further analysis of the differentiation potential of our cells under these conditions and any transcriptional changes. This is beyond the scope of the current manuscript. However, we do think it is an intriguing question that should be addressed in future work.

2. Gastruloid experiments:

- The authors show that cilia appear in areas of the gastruloids that have lumens. What is the in vivo equivalence of these lumens? What is the identity of the cells that surround

these lumens? As far as I know, gastruloids do not have an equivalent of the pro-amniotic cavity.

We thank the reviewer for this question. The short answer is that we do not have a clear idea of what these lumens are. They most likely represent localized, tissue-specific epithelial cavities similar to the transient lumens seen during organogenesis *in vivo*. We have now included a discussion of their nature and the relevant references in the revised manuscript. A closer inspection of these lumens and their identity is planned in future work, but is beyond the scope for the current manuscript.

- What is the % of cells with cilia at each step of the gastruloid protocol? From the data, it appears that only a small % of cells have cilia after the CHIR pulse. If this is the case, then the initial pool of pluripotent cells before the pulse (equivalent of E5.5 epiblast) does not have cilia, which would not mimic what happens in the embryo. Could it be that the lack of effects observed in terms of differentiation and lineage specification are a consequence of the lack of cilia in the primed pluripotent cells in this model? If this is the case, then alternative in vitro models could be considered. Rosettes can be used as a starting point to study early primitive streak differentiation (Harrison et al, Science, 2017, and Satoh et al, Developmental Cell, 2024). This should be discussed.

After the Chiron pulse at 72 h, over 60% of centriole pairs form a cilium. The number of cells with cilia is therefore very high. Before the pulse, the number is substantially lower (see Figure 5 C). Our current rosette protocol is unfortunately not compatible with further differentiation, and requires adaptation to facilitate the study of eg primitive streak differentiation. We thank the reviewer for this suggestion and will incorporate it into our future work.

Minor comments:

- Figure 1A: The drawing of the blastocyst is not correct; there are too many epiblast cells.
The reviewer is correct. We have updated Figure 1 A to correct this imbalance.

September 30, 2025

RE: Life Science Alliance Manuscript #LSA-2025-03358-TR

Dr. Christa Buecker
Max Perutz Labs
Max Perutz Labs
Dr Bohrgasse 9
Vienna 1030
Austria

Dear Dr. Buecker,

Thank you for submitting your revised manuscript entitled "In vitro approaches to study centriole and cilium function in early mouse embryogenesis".

Your revised manuscript was evaluated by all of the original reviewers whose comments are appended below. We would be happy to publish your paper in Life Science Alliance pending final revisions necessary to meet our formatting guidelines.

- We thank you for providing details for all reagents used in the study. Please refer to this file in the methods section.
- Please state the source of the mESC line in the methods section, or in the associated excel sheet.
- Please provide the primers used for the qPCR experiments including that of the reference gene.
- Please remove the claim on page 12 supported by "data not shown". Alternatively you can include the evidence that is referenced instead of "data not shown".
- Some images appear to be repeated in different figures. We ask you to check this. If the images are indeed the same, then it must be clearly stated in both figure legends that the images are used in the other figure. These are:
Figure S4A (96h) and Figure S7D (96h),
Figure S5B (WT, TP53KO) and Figure 6A (WT, TP53KO),
Figure 3C (WT) and Figure 7D (WT)
Figure 4A (WT) and Figure 8E (WT)
Figure 5B (WT) and Figure 8C (WT)
- Please upload your main manuscript text as an editable doc file.
- Please upload all figure files as individual ones, including the supplementary figure files.
- Please add ORCID ID for the corresponding author - you should have received instructions on how to do so.
- Please add a Category for your manuscript in our system.
- Please add the X and Bluesky handles of your host institute/organisation, as well as your own and/or one of the authors, in our system.
- Please clearly mark the corresponding author on the title page of the manuscript file.
- The "Data Availability" section should be placed after the Materials & Methods section. Please consult our guidelines at <https://www.life-science-alliance.org/manuscript-prep#format>
- Please add a Conflict of Interest statement to your main manuscript text.
- Please use the [10 author names, et al.] format in your references (i.e., limit the author names to the first 10).
- Please add your main and supplementary figure legends to the main manuscript text after the references section.
- It is recommended to exclude figures from the manuscript text and upload them separately.
- We encourage you to revise the figure legend for Figure S3 such that the figure panels are introduced in alphabetical order.
- Please be sure that the authorship listing and order is correct.

A. FINAL FILES:

B. MANUSCRIPT ORGANIZATION AND FORMATTING:

Thank you for your attention to these final processing requirements. Please revise and format the manuscript and upload materials as soon as you are able.

Sincerely,

Sarita Hebbar, PhD
Scientific Editor
Life Science Alliance
<http://www.lsjournal.org>

Reviewer #1 (Comments to the Authors (Required)):

The revised manuscript is significantly improved; I support its publication.

Reviewer #2 (Comments to the Authors (Required)):

In the revised manuscript, the authors have addressed all my concerns well.

Reviewer #3 (Comments to the Authors (Required)):

The authors have provided a discussion for the points that I raised in my revision. I am satisfied with the current version and support its publication.

October 13, 2025

RE: Life Science Alliance Manuscript #LSA-2025-03358-TRR

Dr. Christa Buecker
Max Perutz Labs
Max Perutz Labs
Dr Bohrgasse 9
Vienna 1030
Austria

Dear Dr. Buecker,

Thank you for submitting your Research Article entitled "In vitro approaches to study centriole and cilium function in early mouse embryogenesis". It is a pleasure to let you know that your manuscript is now accepted for publication in Life Science Alliance. Congratulations on this interesting work.

DISTRIBUTION OF MATERIALS:

Again, congratulations on a very nice paper. I hope you found the review process to be constructive and are pleased with how the manuscript was handled editorially. We look forward to future exciting submissions from your lab.

Sincerely,

Sarita Hebbar, PhD
Scientific Editor
Life Science Alliance
<http://www.lsajournal.org>